# LooongLLaVA: Scaling Multi-modal LLMs to 1000 Images Efficiently via a Hybrid Architecture

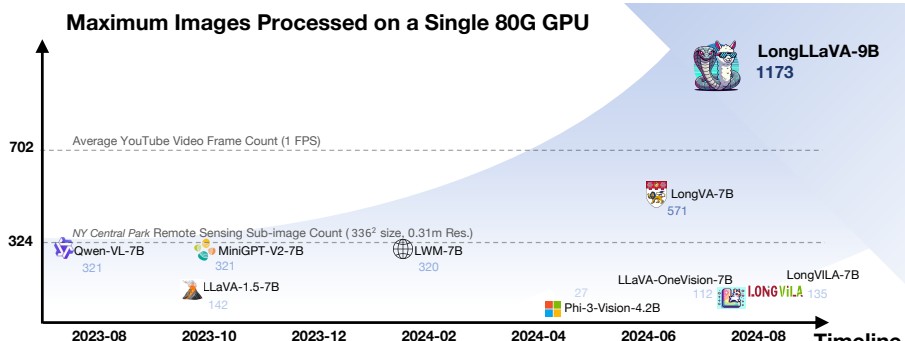

Figure 1: Comparison of the maximum images processed by MLLMs on a single 80GB GPU (`Int8` Quantization), and plotted against their release dates. Our model, LongLLaVA, leads the way with the ability to handle up to 1173 images, demonstrating its superior processing capability. Res refers to resolution. Although these baseline models are capable of processing these images as input, their performance often deteriorates significantly (Song et al., 2024) with more images.

## Abstract

Expanding the long-context capabilities of Multi-modal Large Language Models (MLLMs) is crucial for video understanding, high-resolution image understanding, and multi-modal agents. This involves a series of systematic optimizations, including model architecture, data construction and training strategy, particularly addressing challenges such as *degraded performance with more images* and *high computational costs*. In this paper, we adapt the model architecture to a hybrid of Mamba and Transformer blocks, approach data construction with both temporal and spatial dependencies among multiple images and employ a progressive training strategy. The released model **LongLLaVA** (**Long**-Context **L**arge **L**anguage **a**nd **V**ision **A**ssistant) is the first hybrid MLLM, which achieved a better balance between efficiency and effectiveness. LongLLaVA not only achieves competitive results across various benchmarks, but also maintains high throughput and low memory consumption. Especially, it could process nearly a thousand images on a single A100 80GB GPU, showing promising application prospects for a wide range of tasks.

## 1 Introduction

The rapid advancement of MLLMs (Liu et al., 2024b; 2023a; Dong et al., 2024a; Chen et al., 2024a) has demonstrated their remarkable capabilities across various applications (Chu et al., 2024; Yang et al., 2023; Wu et al., 2023b; Chen et al., 2024b). However, multi-image scenario remain an important yet to-be-explored aspect. In particular, expanding the context of MLLMs to understand longer videos (Zhang et al., 2023; Cheng et al., 2024b), higher-resolution images (Xu et al., 2024c; Wu & Xie, 2023b), and make decisions based on more historical messages (Wang et al., 2024b; Liu et al.,

2024c) is crucial for enhancing user experience (Li et al., 2024b) and further broadening MLLMs' application scope (Apple, 2024).

However, extending the context length of MLLMs to improve their usability poses challenges related to degraded performance and high computational costs when processing more images. To maintain the performance in longer context, some studies (Zhang et al., 2024a; Zhao et al., 2024c) have concentrated on curating long-context training data involving multiple images to enhance performance. Additionally, other research efforts have explored innovative training strategies (Liu et al., 2024a; Zhang et al., 2024b; Li et al., 2024a; Zhang et al., 2024d) to mitigate performance declines. Regarding the issue of high computational costs, Xue et al. (2024) have made strides in improving multi-node efficiency by reducing communication costs. However, there remains a gap in solutions for accelerating the computation itself when managing longer contexts.

To address the challenges mentioned above, we propose a systematic solution called **LongLLaVA**, especially using a hybrid architecture for acceleration. This solution comprehensively optimizes across three dimensions: *Multi-modal Architecture*, *Data Construction*, and *Training Strategy*.

- For **Multi-modal Architecture**, we adopt a hybrid hybrid Transformer-Mamba architecture and an efficient image representation method that applies 2D pooling to compress image tokens, significantly reducing computational costs while maintaining performance.

- For **Data Construction**, we have designed unique formats for different tasks, enabling the model to distinguish between temporal and spatial dependencies among images.

- For **Training Strategy**, we use a three-stage method for multi-modal adaptation—Single-image Alignment, Single-image Instruction-tuning, and Multi-image Instruction-tuning—to incrementally enhance the model's ability to handle multi-modal long contexts.

Experiemntal results show that LongLLaVA excels in understanding multi-modal long contexts with high efficiency. It leads in retrieval, counting, and ordering tasks in VNBench (Zhao et al., 2024e) and achieves nearly 100% accuracy with 1,000 images on a single 80GB GPU for Needle-In-A-Haystack evaluation (Zhang et al., 2024b). Our summarized contributions are as follows:

- We introduce LongLLaVA, a solution optimized through data construction, training strategies, and multi-modal architecture, effectively balancing performance and efficiency. To the best of our knowledge, this is the first hybrid architecture for MLLMs.

- LongLLaVA demonstrates exceptional performance in multi-modal long-context understanding, excelling in retrieval, counting, and ordering tasks. In our commitment to transparency and community research, we will open source all models, codes, and datasets associated with LongLLaVA.

## 2 TOWARDS SCALING UP THE IMAGE NUMBER IN MLLMS

### 2.1 THE CURSE OF IMAGE NUMBERS

**Degraded Performance with More Images.** While many open-source MLLMs match closed-source models on single-image tasks (Bai et al., 2023; Li et al., 2024a; Zhang et al., 2024a; OpenAI, 2024; Google, 2024), their performance degrades significantly in multi-image scenarios, particularly in tasks involving temporal or semantic relationships (Song et al., 2024). This limitation restricts their usability and calls for systematic solutions from the open-source community.

**Excessive Input Length.** Processing multiple images results in excessive input length due to the large number of tokens generated by visual encoders like CLIP (Radford et al., 2021). For example, representing a three-minute video at 1 FPS requires 103,680 tokens, causing increased computational demand and memory usage. Compression methods (Chen et al., 2023a; Zhang et al., 2024b; Xu et al., 2024b) partially alleviate this issue but often compromise performance.

**High Computational and Memory Complexity.** The quadratic scaling of Transformer architectures with sequence length leads to high computational and memory overhead when handling multiple images. Techniques like ring attention (Liu et al., 2024a; Zhang et al., 2024b), sequence parallelism (Xue et al., 2024), and Mamba architectures (Gu & Dao, 2024; Zhao et al., 2024a) offer

partial relief but introduce trade-offs, such as time overhead or reduced in-context learning capabilities (Lieber et al., 2024). A balanced solution is needed to address these challenges in multimodal contexts.

## 2.2 MOTIVATION FOR HYBRID ARCHITECTURE

Table 1: Comparative Analysis of Architectures. A checkmark (✓) indicates that the architecture supports in-context learning (ICL) capabilities, while a cross (✗) denotes a relatively weaker ICL ability. For a more detailed experimental analysis, please refer to Sec. 5.1.

| Architecture | Compute Complexity | ICL | Representative models |
|---|---|---|---|
| Transformer | Quadratic | ✓ | Gemma (Team et al., 2024), LLaMA (Touvron et al., 2023) |
| Mamba | Linear | ✗ | Mamba (Gu & Dao, 2024), Mamba-2 (Dao & Gu, 2024) |
| Hybrid | Quasi-Linear | ✓ | Jamba (Lieber et al., 2024), Zamba (Glorioso et al., 2024) |

Transformer architectures are highly effective in multimodal tasks but face significant computational challenges due to their quadratic complexity with sequence length. This inefficiency becomes a bottleneck in long-context scenarios, requiring high memory and computation resources. Mamba architectures address this issue with their linear computational complexity, making them significantly more efficient. However, they exhibit notable weaknesses in In-Context Learning (ICL) tasks, particularly those involving complex retrieval or reasoning (Park et al., 2024). These limitations may attributed to Mamba's reliance on reduced attention mechanisms (Olsson et al., 2022), which constrain its ability to learn contextual patterns effectively. While explicit training can enable Mamba models to perform simple ICL tasks, this approach restricts the utilization of the model's full capacity and training data (Dao & Gu, 2024).

Recent advancements have demonstrated the potential of hybrid Mamba-Transformer architectures, which integrate Mamba's efficiency with the robust ICL capabilities of Transformers (Dao & Gu, 2024; Wang et al., 2024a). Comparative experiments show that these hybrids achieve superior performance on ICL tasks and maintain computational efficiency. For instance, Jamba (Lieber et al., 2024), a hybrid model, can process 256K tokens with only 4GB of KV-Cache memory, far surpassing the capabilities of Mixtral (Jiang et al., 2024a), which has the same activation parameters. This balance between effectiveness and efficiency makes hybrid architectures an ideal solution for long-context multimodal tasks, addressing both computational and functional limitations.

## 2.3 THE BENEFIT OF SCALING UP THE IMAGE NUMBER

Adopting more images significantly broadens the application scenarios for current MLLMs. We will explore this from two dimensions: *Temporal Expansion* and *Spatial Expansion*.

**Temporal Expansion.** Understanding the temporal dependencies between images is crucial for a variety of applications. In multi-modal assistants, it enhances real-time recall capabilities, which is particularly beneficial for the elderly (Li et al., 2024b; Loveys et al., 2022). For mobile agents, it enables more personalized services and improves task planning (Deng et al., 2024; Li et al., 2024f; Wu et al., 2023a). In the healthcare sector, it assists in anomaly detection in 3D medical videos, thereby reducing diagnostic errors (Bai et al., 2024a).

**Spatial Expansion.** When dealing with high-resolution images (Xu et al., 2024c; Dong et al., 2024b) or when detailed understanding of images (Wu & Xie, 2023b) is required, images are often decomposed into sub-images. This process highlights the importance of grasping spatial dependencies among these sub-images. In remote sensing, an increased number of images enhances both coverage and granularity (Guo et al., 2024; Liu et al., 2022). In pathology, it minimizes information loss and improves diagnostic accuracy (Sun et al., 2024; Xu et al., 2024a). In the field of Molecular Learning, it facilitates the processing of complex reactions and the analysis of larger molecular graphs (Zhang et al., 2024c; Le et al., 2024).

# 3 LONGLLAVA: SCALING LLAVA TO LONGER CONTEXT

To address the aforementioned challenges and enhance the model's adaptability to long-context, multi-image scenarios, we introduce improvements from three perspectives: *multi-modal model architecture* (Sec. 3.1), *data processing protocol* (Sec. 3.2), and *training strategy* (Sec. 3.3).

## 3.1 MULTI-MODAL ARCHITECTURE

Our multimodal architecture is constructed around three core components inspired by LLaVA (Li et al., 2024a): the Vision Encoder, the Projector, and the LLM. The primary strategies for adapting to multimodal long-context are predominantly derived from two aspects.

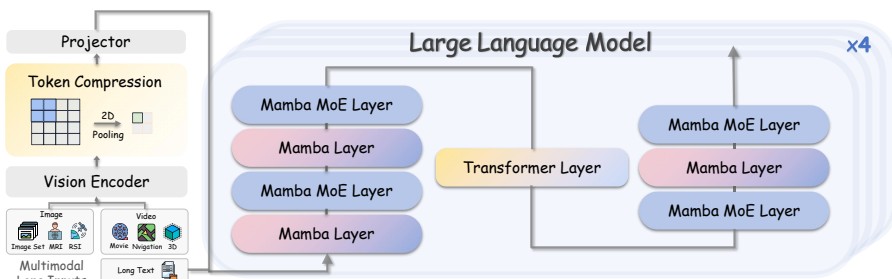

Figure 2: **Architecture of LongLLaVA.** The LongLLaVA model is capable of (1) accommodating a variety of multimodal inputs and efficiently processing image tokens via 2D token compression; (2) uniformly managing the preprocessed inputs within its hybrid LLM architecture, which comprises four stacks of hybrid layers, each blending Transformer and Mamba layers in a 7:1 ratio.

**Vision Information Processing.** We employ CLIP[1] as the vision encoder to encode visual information and a two-layer MLP as the projector to map vision features into the text embedding space suitable for the LLM. Prior to projection, bilinear pooling is applied, reducing the token representation of an image from 576 to 144 by aggregating $2 \times 2$ patch units into a single token. This approach effectively conserves training and inference time while maintaining essential spatial relationships between patches. Further details on the effectiveness of this strategy are provided in Section 4.5.

**Hybrid LLM Architecture.** Our model employs a hybrid LLM architecture comprising four stacks of hybrid layers, each integrates Transformer and Mamba layers in a 7:1 ratio, as depicted in Figure 2. It also features a Mixture of Experts (MoE) approach in every other layer, utilizing 16 experts and selecting the top-2 experts for each token. RMSNorm (Zhang & Sennrich, 2019) is used between layers to enhance normalization, although positional embeddings are omitted. The model incorporates Grouped Query Attention (GQA) (Ainslie et al., 2023) and SwiGLU activation functions (Shazeer, 2020), similar to other large language models. The total parameter count of the model is 53B, with activation parameters during inference totaling 13B; we designate this model as **LongLLaVA-A13B**. In an effort to make the model more efficient, we have retained only the Expert-0 in the Mamba MoE Layer[2], thereby constructing **LongLLaVA-9B**.

## 3.2 DATA PROCESSING PROTOCOL

To ensure that the model effectively distinguishes between temporal and spatial dependencies among images in multi-image scenarios and performs well across various tasks, we meticulously differentiated special characters in different scenarios. As shown in Figure 3, these special characters comprehensively address the various relationships between images in different contexts, thereby enhancing the model's adaptability to diverse tasks.

**Regular Single and Multiple Images.** For this type of inputs, we use  and </img> to enclose image tokens, helping the model differentiate between image and text tokens.

**Video.** For video inputs, to enable the model to understand the temporal relationship between frames, we first use <vid> and </vid> to enclose image tokens. Additionally, we add the special symbol <t> between different frames to represent the temporal dependency between them.

**High Resolution Image.** For complex single-image understanding that require dividing an image into multiple sub-images, we use \n to separate the main image from its sub-images. For the arrangement of sub-images, we traverse from the top-left to the bottom-right, adding \n between split lines to preserve the relative spatial positions of the sub-images.

---

[1] openai/clip-vit-base-patch32
[2] We chose Expert-0 due to minimal performance differences, detailed in Appendix A.

> **Data Processing Protocol**
>
> **In the Following Statement: `<Image>`=`<img_token>...</img>`**
> *For Single-image*: "`<Image>`\n What is this?"
> *For Multi-image*: "`<Image>`\n This is a cat. `<Image>`\nThis is a:"
> *For Video*: "`<vid><Image><t>...<Image></vid>`\n What are they?"
> *For Patched-image*: "`<Image>`\n`<Image>`..\n..`<Image>`\n What are they?"

Figure 3: **Data Processing Protocol for LongLLaVA.** We utilized different tokens to distinguish various modal information, and to identify the spatial and temporal relationships within images.

## 3.3 TRAINING STRATEGY

In our training strategy, we implement single-modal and multi-modal adaptations to transform a pre-trained language model into a multimodal long-context model.

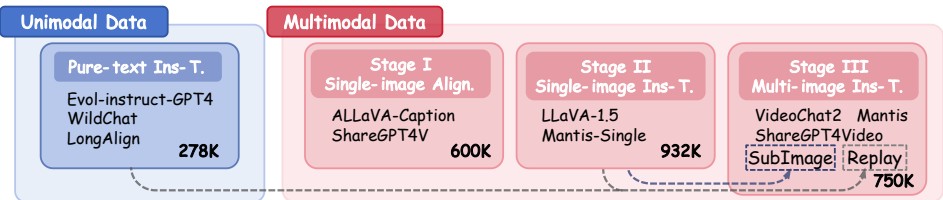

Figure 4: **Dataset Taxonomy of LongLLaVA**. `Replay` refers to data sampled from former phase to maintain single-image and dialogue understanding ability. `SubImage` denotes a constructed dataset for understanding complex single images divided into sub-images. Ins-T. and Align. refer to instruction-tuning and alignment, respectively.

**Pure-text Instruction Tuning.** We initially enhance the pre-trained language model's ability to follow instructions of varying lengths in pure-text contexts. This is achieved using a comprehensive dataset totaling 278k pure text entries from Evol-instruct-GPT4 (Xu et al., 2023), WildChat (Zhao et al., 2024d), and LongAlign (Bai et al., 2024b).

For multi-modal adaptation, following the *Single-image Alignment* and *Single-image Instruction-tuning* stages in LLaVA (Li et al., 2024a), we introduce a *Multi-image Instruction-tuning* stage to progressively enhance the model's long-context capabilities. We adopt progressive training not only for better control of variables but also to increase model reusability (Fu et al., 2024b). The specific dataset usage is detailed in Figure 4.

**Stage I: Single-image Alignment.** This stage is to align visual modal features with textual modality. We utilize datasets such as ALLaVA-Caption (Chen et al., 2024a) and ShareGPT4V (Chen et al., 2023b), which comprise approximately 600K high-quality image-caption pairs. During this phase, only the projector is trained while freezing the parameters of the Visual Encoder and LLM.

**Stage II: Single-image Instruction Tuning.** This stage aims to endow the model with multimodal instruction-following capabilities. We use datasets like LLaVA-1.5 (Liu et al., 2023b) and Mantis-Single (Jiang et al., 2024b), totaling around 932K high-quality question-answer pairs. Here, only the Visual Encoder is frozen, and the projector and LLM parts are trained. This process ultimately results in the development of LongLLaVA (single image).

**Stage III: Multi-image Instruction Tuning.** In this stage, the model is trained to follow instructions in multimodal long-context scenarios. We sample 200K, 200K and 50K data items from Mantis (Jiang et al., 2024b), VideoChat2 (Li et al., 2024d) and ShareGPT4Video (Chen et al., 2024c) respectively. To preserve the model's single-image comprehension and pure-text dialogue capabilities, we include an additional 200K and 50K data items from the Single-image Instruction-tuning and Pure-text Instruction-tuning phases as the `Replay` component. Furthermore, to enhance the model's ability to interpret complex single images segmented into multiple sub-images, we extract 50K data items from the Single-image Instruction-tuning phase, perform padding and segmentation, and divide the original images into sub-images of size $336 \times 336$ as the `Sub-Image` component.

Table 2: Results of Multi-image Evaluation. PFLOPs represents the number of floating-point operations required to infer 128 images. The highest scores for proprietary and open-source MLLMs are marked in bold. Video-MME is evaluated under the settings of without subtitles. Precision is FP16.

| Model | PFLOPs | #P. | MileBench | | | | VideoMME w/o subs | | | | MVBench |
|---|---|---|---|---|---|---|---|---|---|---|---|
| | | | Temporal | Semantic | IR | Avg. | Short | Medium | Long | Avg. | |
| **Proprietary Models** | | | | | | | | | | | |
| GPT-4V | - | - | 45.6 | 58.9 | 86.7 | 63.7 | 70.5 | 55.8 | 53.5 | 59.9 | **43.5** |
| GPT-4o | - | - | **56.2** | **63.5** | **88.8** | **69.5** | 72.5 | 63.1 | 58.6 | 64.7 | - |
| Gemini-1.5-Pro | - | - | 50.2 | 58.3 | 88.0 | 65.5 | **78.8** | **68.8** | **61.1** | **69.6** | - |
| **Open-source MLLMs** | | | | | | | | | | | |
| Video-LLaMA2 | 3.71 | 7B | - | - | - | - | 55.9 | 45.4 | 42.1 | 47.8 | 34.1 |
| VideoChat2 | 0.24 | 7B | 25.5 | 25.5 | 9.2 | 20.1 | 48.3 | 37.0 | 33.2 | 39.5 | 51.9 |
| LongVILA | 3.90 | - | 8B | - | - | - | 61.8 | 49.7 | 39.7 | 50.5 | - |
| LongVA | 4.90 | - | 8B | - | - | - | 61.1 | 50.4 | 46.2 | 52.6 | - |
| Phi-3-Vision | 2.68 | 3.8B | 46.9 | 50.0 | 18.7 | 38.5 | - | - | - | - | - |
| OmChat | 3.90 | 8B | 51.4 | 52.0 | 34.2 | 45.9 | - | - | - | - | 50.2 |
| **LongLLaVA-9B** | 0.15 | 9B | 48.6 | 47.6 | 48.2 | 48.1 | 54.2 | 44.1 | 38.2 | 45.5 | 50.2 |
| *+ More Data** | 0.15 | 9B | 52.2 | 51.4 | 52.8 | 52.1 | 58.4 | 48.3 | 41.7 | 49.5 | 54.2 |
| **LongLLaVA-A13B** | 0.22 | 53B | **54.1** | **55.0** | **68.5** | **59.2** | **62.9** | **52.2** | **46.4** | **53.8** | **56.2** |

[*] 'More Data' indicates model trained for around two epochs (82 hours $\times$ 8 $\times$ A800-80G) to ensure comparable training time to LongVA (84 hours $\times$ 8 $\times$ A100-80G).

## 4 EXPERIMENTS

### 4.1 TRAINING DETAILS

For training, we utilize random sampling to concatenate data items into a token length of 176,000, separated by the `<eos>` token. This approach helps in managing extensive datasets and ensuring diverse coverage of different data segments. Training is executed across three compute nodes, each equipped with eight A800 GPUs, leveraging DeepSpeed Zero-3 as the distributed strategy to enhance scalability and efficiency. We employ a cosine learning rate scheduler with a warm-up rate of `0.03`, set the training epoch to 1, and the learning rate to `1e-5`.

### 4.2 EVALUATION SETUP

**Benchmarks.** We mainly focus on evaluating the model's multimodal long-context understanding ability using three multi-image benchmarks: MileBench (Song et al., 2024) for assessing multimodal long-context scenario performance, and Video-MME (Fu et al., 2024a) along with MVBench (Li et al., 2024d) for video analysis capabilities. Detailed descriptions of these benchmarks are available in Appendix B. For basic single-image evaluations, please refer to Appendix C for details.

**Models.** We compare our model against four commercial models: GPT-4V[3] (OpenAI, 2024), GPT-4o[4], Gemini-1.5-Pro[5] (Google, 2024), and Claude3-Opus[6], as well as five open-source models: Phi-3-Vision[7], OmChat (Zhao et al., 2024b), LongVA, LongVILA (Xue et al., 2024), Video-LLaMA-2 (Cheng et al., 2024a) and VideoChat2 (Li et al., 2024d). Additionally, the temperature is set to zero to guarantee consistent performance evaluation. Unless specified otherwise, LongLLaVA-9B and LongLLaVA-A13B are evaluated using `Int8` quantization, a method designed to reduce computational costs while preserving performance and "LongLLaVA" refers to LongLLaVA-A13B.

### 4.3 MAIN RESULTS

As shown in Table 2, LongLLaVA demonstrates superior performance among open-source models on MileBench, even surpassing Claude3-Opus, and particularly excels in retrieval tasks. This highlights LongLLaVA's impressive capabilities in handling multi-image tasks. Notably, LongLLaVA's effectiveness is further underscored by its performance on video benchmarks such as Video-MME and MVBench with an order of magnitude fewer FLOPs. It shows exceptional results, especially in tasks involving medium to long-length videos, outperforming traditional video models like Video-LLaMA2 and VideoChat2.

---

[3] `gpt-4-vision-preview`
[4] `https://openai.com/index/hello-gpt-4o/`
[5] `gemini-1.5-pro`
[6] `claude-3-opus-20240229`
[7] `https://huggingface.co/microsoft/Phi-3-vision-128k-instruct`

## 4.4 DIAGNOSTIC EVALUATION OF LONG-CONTEXT MLLMS

Table 3: Long Context MLLMs' Atomic Capabilities Analysis using VNBench (Zhao et al., 2024e). PFLOPs refers to the number of floating-point operations required for inference on 54 images, which corresponds to the average number of frames extracted from the dataset videos at 1 FPS.

| Video MLLM | PFLOPs | #P | Retrieval | | | Ordering | | | Counting | | | Avg. |
|---|---|---|---|---|---|---|---|---|---|---|---|---|
| | | | E | I-1 | I-2 | E | I-1 | I-2 | E-1 | E-2 | I | |
| **Proprietary Models** | | | | | | | | | | | | |
| Gemini-1.5 | - | - | 100.0 | 96.0 | 76.0 | 90.7 | 95.3 | 32.7 | 60.7 | 7.3 | 42.0 | 66.7 |
| GPT-4o | - | - | 100.0 | 98.0 | 87.3 | 88.4 | 86.6 | 45.2 | 36.8 | 0.0 | 36.1 | 64.4 |
| GPT-4V | - | - | 100.0 | 99.3 | 82.0 | 42.6 | 22.8 | 23.0 | 37.6 | 0.0 | 32.4 | 48.9 |
| **Open-source MLLMs** | | | | | | | | | | | | |
| Video-LLama2 | 0.85 | 7B | 1.2 | 26.0 | 6.0 | 0.0 | 0.0 | 0.0 | 2.0 | 4.7 | 0.7 | 4.5 |
| VideoChat2 | 0.08 | 7B | 43.4 | 40.0 | 14.6 | 0.0 | 0.0 | 1.3 | 4.4 | 8.0 | 12.4 | 12.4 |
| **LongLLaVA-9B** | 0.07 | 9B | 98.3 | 57.2 | 96.3 | 24.2 | 57.2 | 24.3 | 24.5 | 21.0 | 26.0 | 44.4 |
| **LongLLaVA-A13** | 0.09 | 53B | **100** | **73.3** | **100.0** | **37.5** | **35.3** | **34.8** | **36.0** | **23.7** | **28.0** | **52.1** |

Considering that former evaluations cannot adequately capture the abilities of MLLMs over long contexts, we use a diagnostic evaluation set, VNBench (Zhao et al., 2024e), to further analyze the atomic capabilities of models in long contexts. VNBench is a benchmark construction framework based on synthetic video generation, encompassing tasks such as retrieval, ordering, and counting.

The results, as presented in Table 3, indicate that LongLLaVA exhibits performance that is on par with leading closed-source models in tasks such as cross-context retrieval, ordering, and technical capabilities, even outperforms GPT-4V. Among open-source models, LongLLaVA also shows its superior performance. This positions LongLLaVA as a prominent contender in the field, demonstrating its advanced capabilities in managing and interpreting long contexts. To further assess the retrieval ability of LongLLaVA, we conducted multimodal needle-in-a-haystack experiment, the specifics of which are outlined in Appendix D.

## 4.5 ABLATION STUDY

As shown in Table 4, significant improvements were observed across all evaluation sets when using the **hybrid LLM architecture**, Jamba, with identical data and model parameters, demonstrating its potential in multimodal scenarios. For **token compression**, we choose the 2D pooling, which significantly reduces computational load while keeping performance degradation within acceptable limits (less than 2.2%). Compared to 1D pooling, the 2D pooling method, which pools along the height and width di-rections to obtain a 12x12 token arrangement, yields better results (0.1∼1.5 improvement). For **training strategy**, the results indicate that progressive training achieves better performance on multi-image tasks while maintaining comparable results on single-image tasks. For more ablation studies on token number per image, LLM architecture and replay data and , please see Appendices E.1, E.2 and E.3 for details.

Table 4: Ablation on ▓ model architecture, ▓ dataset construction and ▓ training strategy. Each strategy builds upon the previous row, except for 1D Pooling. $Mile^*_{avg}$ is the average score of MileBench. 1D and 2D denote different pooling strategies. #T refers to the token count for one image. & refers to the combination of the stages.

| Method | #T | GQA | MMMU | $SQA^I$ | $SEED^{v1}_{img}$ | $Mile^*_{avg}$ |
|---|---|---|---|---|---|---|
| **Architecture & Data Abalation on LongLLaVA-A13B** | | | | | | |
| LLaVA-1.5-13B | 576 | 63.3 | 34.4 | 71.6 | 68.2 | 27.6 |
| +Jamba as LLM | 576 | 63.2 | 41.4 | 75.4 | 69.8 | 38.2 |
| *+1D Pooling* | 144 | 60.4 | 42.0 | 73.9 | 66.3 | 36.2 |
| +2D Pooling | 144 | 61.3 | 42.1 | 75.2 | 67.4 | 37.7 |
| +Single-image Data | 144 | 62.2 | 42.1 | 75.9 | 68.9 | 50.0 |
| +Multi-image Data | 144 | 59.9 | 39.2 | 73.4 | 65.3 | 57.4 |
| **Training Strategy Abalation on LongLLaVA-9B** | | | | | | |
| *+Stage1&2&3* | 144 | 56.9 | 32.8 | 67.2 | 66.9 | 42.2 |
| *+Stage1, 2&3* | 144 | 57.6 | 33.2 | 70.2 | 68.4 | 44.2 |
| +Stage1, 2, 3 | 144 | 58.4 | 34.4 | 69.9 | 67.9 | 46.5 |

## 5 MORE ANALYSIS

In this section, we conduct further analysis to understand the inner workings and multimodal long-context capability of LongLLaVA.

## 5.1 ON THE MOTIVATION FOR THE HYBRID ARCHITECTURE

Table 5: ICL Capability and Efficiency Analysis across Different Architectures. TP and Mem. refer to throughput and memory usage.

| Model | Arch. | Active Param. | #Few-shot of VL-ICL | | | | 100K Token (Efficiency) | | | |
|---|---|---|---|---|---|---|---|---|---|---|
| | | | 1 | 2 | 4 | 5 | Prefill (s) | TP (tokens/s) | Mem. (GB) | Max TP (tokens/s) |
| Falcon-mamba | Mamba | 7B | 49.0 | 51.9 | 52.4 | 53.2 | 24.3 | 32.4 | 48.8 | 90.7 |
| LLaVA-1.6 | Transformer | 13B | 50.0 | 52.3 | 54.6 | 58.9 | 34.0 | 14.7 | 79.4 | 14.7 |
| **LongLLaVA-9B** | Hybrid | 9B | 51.6 | 57.8 | 58.4 | 60.2 | 16.5 | 62.1 | 38.7 | 155.2 |
| **LongLLaVA-A13B** | Hybrid | 13B | 52.3 | 59.0 | 59.0 | 61.3 | 25.5 | 37.6 | 79.1 | 37.6 |

We explore the strengths and weaknesses of different architectures in terms of ICL capabilities and inference efficiency, highlighting the balanced advantages of multimodal hybrid architectures that combine the strengths of both. For Mamba Architecure, we train and evaluate the Falcon-mamba (Zuo et al., 2024) model with 7.3B parameters using the same settings as our model, as it represents the largest available Mamba configuration, despite the difficulty in aligning parameter counts in MLLMs. For Transformer, we choose the 13B parameter LLaVA-1.6, which has inference parameters consistent with LongLLaVA, to enable a more accurate efficiency comparison.

**ICL Analysis.** We evaluate the performance on the Matching Image task from VL-ICL benchmark (Zong et al., 2024) for multi-modal in-context learning. This task's inputs contain an image pair $x = \{x1, x2\}$, and output $y$ indicates whether a specific relation $r$ holds between them. MLLMs are required to learn the relation from examples. As shown in the Table 5, both Hybrid and Transformer architectures exhibit rapid performance improvements with the increase in examples, whereas the Mamba architecture shows a slower improvement, confirming its ICL shortcomings. Building on the concept of many-shot fine-tuning (Agarwal et al., 2024), we further investigated the inference scalability of the hybrid architecture model with respect to the larger number of ICL shots as Application detailed in Section 6.3

**Efficiency Analysis.** We focus on four aspects: Prefill Time (first inference latency), Throughput (next tokens per second), Memory Usage and Maximum Throughput (throughput under maximum batch size). We control the input text length to 100K and measure time and maximum memory usage for generating outputs of 1 token and 1000 tokens. Throughput is calculated as $(1000 - 1)/(time_{1000} - time_1)$. To better simulate real application scenario, Transformer and Hybrid architectures are evaluated using vLLM framework (Kwon et al., 2023) with `Int8` quantization (Frantar et al., 2023). As shown in the Table 5 the Hybrid architecture achieves 2.5 times the Throughput, 4.1 times the Maximum Throughput, 75% of the Prefill Time, and reduced memory usage compared to the Transformer architecture with similar inference parameters.

## 5.2 SCALING LAW OF THE IMAGE NUMBER

With more images processed, it could support more image patches for high-resolution image understanding and more video frames for video understanding. To explore the impact of increasing the number of sub-images and video frames, we evaluate LongLLaVA on the benchmarks V* Bench (Wu & Xie, 2023a) and Video-MME (Fu et al., 2024a) respectively.

**Scale up Number of SubImages.** V* Bench evaluates a model's ability to locate small objects within large images. As shown in Figure 5, increasing the number of sub-images initially improves model performance significantly, indicating better understanding of image details. However, we also find that further increasing the number of sub-images slightly degraded the performance, suggesting that an excessive number of sub-images may interfere with performance on this task.

**Scale up Number of Frames.** Video-MME (Fu et al., 2024a) is a benchmark that tests a model's ability to extract information from videos. We can see from Figure 6 that as the number of sampled frames increases, the model's performance on the benchmark improves significantly, reaching its peak when 256 frames are extracted. This indicates that the model can effectively understand and utilize the information contained in the additionally sampled frames to provide better responses.

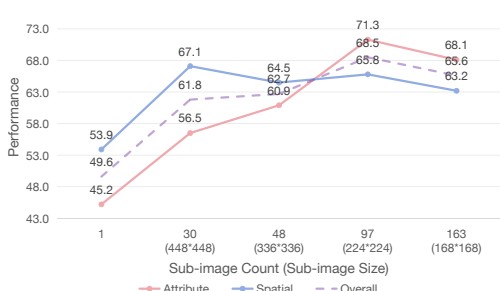 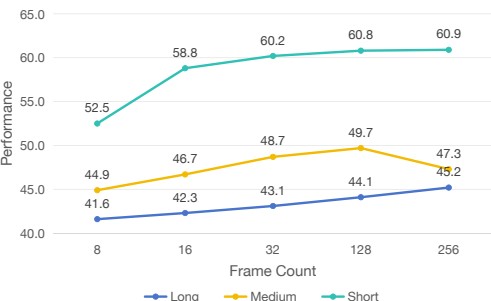

Figure 5: Performance of LongLLaVA with increasing sub-image counts on V*

Figure 6: Performance of LongLLaVA with increasing frame counts on Video-MME

# 6 MORE APPLICATIONS FOR LONGLLAVA

Apart from the long video understanding task introduced in Section 4.3, which demands a prolonged temporal image comprehension ability, we have investigated three additional domains, Healthcare, Science and Many-shot ICL. These areas necessitate the utilization of LongLLaVA's fine-grained image understanding capability and multimodal long-context understanding ability, providing us a broader platform to further probe its potential applications.

## 6.1 APPLICATION IN HEALTHCARE

Table 6: Performance of the models on the pathology image understanding tasks.

| Model | Size | VQA-RAD | PathVQA |
|-------|------|---------|---------|
| GPT-4V | - | 39.5 | - |
| LLaVA | 34B | 58.6 | 59.1 |
| LLaVA-Med | 7B | 55.5 | 35.9 |
| HuatuoGPT-V | 8B | 63.8 | **59.9** |
| LongLLaVA-Med | 9B | **68.5** | 55.0 |

Table 7: Performance on 3D CT image understanding task. Acc., Rec. and Prec. refer to Accuracy, Recall and Precision, respectively.

| Model | Acc. | Rec. | Prec. | F1 |
|-------|------|------|-------|-----|
| CT-CLIP | 65.1 | 73.8 | 30.4 | 43.0 |
| LongLLaVA-Med | **86.7** | **77.6** | **35.5** | **48.5** |

To evaluate the potential of LongLLaVA in the Healthcare domain, we selected two tasks: Pathology Image Understanding and 3D CT Image Understanding. Initially, we trained LongLLaVA-9B for one epoch using PubMedVision to equip it with basic multimodal medical capabilities, a process which took five hours with eight A800 GPUs. As a result, we obtained LongLLaVA-Med.

**Pathology Image Understanding.** We chose pathological image understanding task, which requires the model to possess fine-grained recognition capabilities and medical knowledge. We evaluated LongLLaVA using two benchmarks, VQA-RAD (Lau et al., 2018) and PathVQA (He et al., 2020). As presented in Table 6, our model proves competitive in comparison to SOTA models (LLaVA-Med (Li et al., 2024c) and HuatuoGPT-V (Chen et al., 2024b)) with less training data.

**3D CT Image Understanding.** To evaluate LongLLaVA's capability in 3D vision tasks, we selected the CT image understanding task. Since 3D CT images can be viewed as a combination of multiple slices of the human body, all slices were converted to RGB format and processed as a multi-image sequence for model interpretation. We conducted a zero-shot evaluation on the CT-RATE (Hamamci et al., 2024) validation set, with random selection of varying resolutions from the same patients. The dataset contains 1304 samples, with slice resolutions ranging from $512 \times 512$ to $1024 \times 1024$, and an average of 690. The number of slices per sample ranges from a maximum of 984 to a minimum of 100, with an average of 300. Table 7 indicate that LongLLaVA-Med outperforms the SOTA model by 21.6 points in terms of accuracy, establishing a new precedent for 3D CT images understanding.

## 6.2 APPLICATION IN SCIENCE

In science domain, we focus on geology and deal with understanding remote sensing images, which needs models are required to perform Visual Question Answering (VQA) based on high-resolution remote sensing images (Zhou et al., 2024). We followed Sky-SenseGPT (Luo et al., 2024), the latest MLLM in this domain and selected the latest FIT-RSFG-VQA task from their work, which is designed to evaluate a model's fine-grained perception capabilities and instruction-following ability in this domain. As shown in Table 8, LongLLaVA maintains excellent performance among all models. Moreover, it surpasses existing SOTA models after fine-tuned on only 27% of the SkySenseGPT data.

Table 8: FIT-RSFG-VQA

| Model | Size | Acc. |
|---|---|---|
| Zero-shot Setting | | |
| LLaVA1.5-7B | 7B | 58.6 |
| GeoChat | 7B | 53.5 |
| LongLLaVA | 9B | **65.2** |
| Fine-tuned Setting | | |
| SkySenseGPT | 7B | 79.8 |
| LongLLaVA-RS* | 9B | **82.3** |

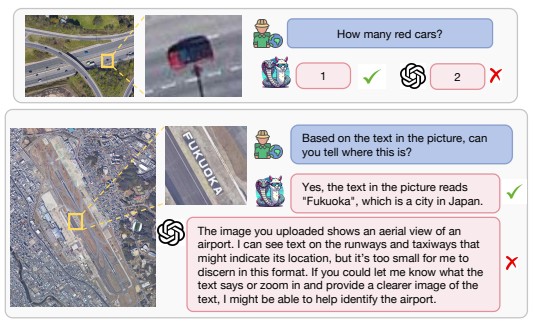

Figure 7: Comparative Study on Remote Sensing

Due to the limited image resolution of FIT-RSFG-VQA, which is only $512 \times 512$, we extended our study by incorporating two high-resolution remote sensing images from the STAR dataset (Li et al., 2024e), with resolutions of $1024 \times 768$ and $3327 \times 4083$, respectively. These images allowed us to conduct a more comprehensive comparative analysis of the model's performance. As shown in Figure 7, LongLLaVA demonstrated its capability to effectively answer VQA questions that required fine-grained recognition. This was achieved by segmenting the images into subimages for processing, resulting in superior performance compared to GPT-4V, particularly in tasks that demanded detailed visual understanding.

## 6.3 APPLICATION IN MANY-SHOT IN-CONTEXT LEARNING.

LLMs often require fine-tuning for optimal performance, but this can be costly and time-consuming, especially when data is scarce or frequent updates are needed. Moreover, fine-tuning might not be feasible in real-world applications with limited resources or rapidly changing tasks. In contrast, many-shot ICL allows models to use more task-specific examples during inference without retraining (Agarwal et al., 2024). To investi-

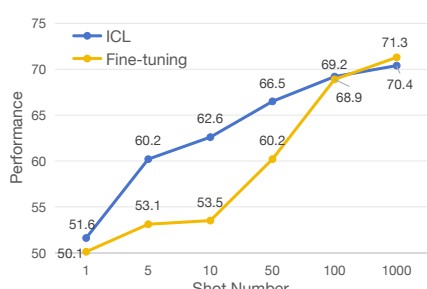

Figure 8: Many-Shot ICL & Fine-Tuning on VL-ICL.

gate the potential of LongLLaVA in many-shot ICL tasks, we compared its performance with different shot counts to fine-tuning on the same number of samples. As shown in Figure 8, LongLLaVA-9B outperforms ICL up to 100 shots. Beyond 1000 shots, the benefit of adding more shots diminishes, and fine-tuning becomes more effective. This suggests that ICL is preferable with fewer than 100 samples, while fine-tuning is more beneficial with around 1000 samples.

## 7 CONCLUSION

In this study, we introduce LongLLaVA, an innovative hybrid architecture model that excels in long-context multi-modal understanding. The model integrates Mamba and Transformer blocks, leveraging temporal and spatial dependencies between multiple images to construct data, and employs a progressive training strategy. LongLLaVA demonstrates competitive performance across various benchmarks while ensuring efficiency, setting a new standard for long-context MLLMs.

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

# Appendix Table of Contents

## A  PRELIMINARY EXPERIMENTS ON EXPERT SELECTION FOR LONGLLAVA-9B

To determine the optimal expert selection method in the MoE layers we also conducted preliminary experiments. Using prevalent LLM benchmarks, MMLU (Hendrycks et al., 2020) and BBH (Suzgun et al., 2022), we evaluated three expert selection strategies: numerical averaging, spherical averaging, and random expert selection.

Table 9: Performance of Different Downcycling Strategies on MMLU and BBH

| Downcycling Strategy | Arithmetic Mean | Spherical Mean | Expert-0 | Expert-5 | Expert-12 | Expert-15 |
|---|---|---|---|---|---|---|
| MMLU | 52.7 | **53.2** | **53.2** | 51.9 | 52.6 | 52.2 |
| Aft. Train | 53.8 | **54.3** | **54.3** | 53.3 | 53.8 | 53.3 |
| BBH | 36.7 | 36.7 | 37.2 | 36.7 | **37.4** | 36.3 |
| Aft. Train | 37.8 | 37.9 | 38.4 | **38.9** | **38.9** | 37.9 |

These methods were compared both before and after Pure-text Instruction Tuning. As shown in Table 9, the differences in model performance were minimal across the selection methods. Therefore, for simplicity, we opted to use Expert-0.

## B  DETAILS OF BENCHMARKS

**Single-image Benchmarks.**  We select seven commonly used evaluations to assess the model's single-image understanding capabilities. These include:

- **GQA** (Hudson & Manning, 2019): A benchmark for real-world visual reasoning and compositional question answering.
- **MME** (Fu et al., 2023): A comprehensive benchmark focused on perception and cognition, from which we use the perception component.
- **MM-Vet** (Yu et al., 2023): Examines six core visual-linguistic (VL) capabilities and sixteen integrations derived from these capabilities.
- **ScienceQA** (Lu et al., 2022): Consists of 4,210 questions across various science topics, with detailed annotations.
- **SEED-Bench-v1** (Li et al., 2023): Evaluates comprehension across twelve dimensions in both image and video modalities; we use the image set.
- **MMBench** (Liu et al., 2023c): A systematically-designed benchmark across twenty ability dimensions.
- **MMMU** (Yue et al., 2024): Tests multi-modal models on multidisciplinary tasks requiring university-level knowledge, covering 183 subfields and 30 types of images.

**Multi-image Benchmarks.**  To explore multi-image capabilities, we utilized:

- **MileBench** (Song et al., 2024): Assesses long-context scenario performance, focusing on Temporal, Semantic, and Information Retrieval (IR) components.
- **Video-MME** (Fu et al., 2024a): Covers 30 sub-fields to evaluate video analysis capabilities. We analyze 128 frames extracted uniformly from each video, independent of subtitles.
- **MVBench** (Li et al., 2024d): Addresses 20 challenging video tasks that are not effectively solved with a single frame.

## C  DETAILS OF SINGLE-IMAGE EVALUATION

The single-image evaluation aims to explore the model's fundamental capabilities and the impact of extended long-context training on single-image understanding.

## C.1 BENCHMARKS

We utilized a series of benchmarks, including GQA (Hudson & Manning, 2019), MME (Fu et al., 2023), MM-Vet (Yu et al., 2023), ScienceQA (Lu et al., 2022), SEED-Bench-v1 (Li et al., 2023), MMBench (Liu et al., 2023c), MMMU (Yue et al., 2024), ChartQA (Masry et al., 2022) and DocVQA (Mathew et al., 2021). These benchmarks assess various aspects of visual understanding and cognitive processing within a single-image context.

## C.2 COMPARISON MODELS

For a comprehensive comparison, we selected three commercial models for the single-image evaluation: GPT-4V[8] (OpenAI, 2024), Gemini-1.5[9] (Google, 2024), and Claude3-Opus[10]. Additionally, we included five open-source models to broaden the scope of evaluation: LLaVA-1.5-13B (Liu et al., 2023b), LLaVA-1.6-13B (Liu et al., 2024b), Phi-3-Vision-4.2B[11] and OmChat-8B (Zhao et al., 2024b).

| Model | TFLOPs | #P | #T | ChartQA | DocVQA | GQA | MM-Vet | $MME^P$ | MMB | MMMU | $SQA^I$ | $SEED^{v1}_{img}$ |
|---|---|---|---|---|---|---|---|---|---|---|---|---|
| **Proprietary Models** | | | | | | | | | | | | |
| GPT-4V | - | - | - | 75.6 | - | - | 67.7 | 1926.5 | **81.3** | **56.8** | 82.1 | 69.1 |
| Gemini-1.5 | - | - | - | **81.3** | **90.9** | - | 65.8 | **2148.9** | 73.6 | 48.9 | 81.4 | 62.9 |
| Claude3-Opus | - | - | - | 80.8 | 89.3 | - | **74.2** | 1586.8 | 63.3 | 54.9 | - | 42.0 |
| **Open-source MLLMs** | | | | | | | | | | | | |
| OmChat | 7.61 | 8B | 576 | - | - | - | 39.6 | - | 78.8 | **45.9** | - | - |
| Phi-3-Vision | 3.56 | 3.8B | 576 | 81.4 | - | - | - | - | **80.5** | 40.4 | **90.8** | - |
| LLaVA-1.6 | 11.86 | 13B | 576 | - | - | **65.4** | **44.9** | 1445.0 | 70.0 | 36.2 | 73.6 | **71.4** |
| LLaVA-1.5 | 11.86 | 13B | 576 | - | - | 63.3 | 36.1 | 1531.1 | 67.7 | 34.4 | 71.6 | 68.2 |
| **LongLLaVA-9B** | 1.04 | 9B | 144 | 44.8 | 47.4 | 58.4 | 32.3 | 1504.6 | 65.6 | 34.4 | 69.9 | 67.9 |
| **LongLLaVA-A13B** | 1.52 | 53B | 144 | 46.3 | 51.2 | 59.9 | 35.2 | 1523.9 | 63.7 | 39.2 | 73.4 | 65.3 |

Table 10: Single-image Evaluation. TFLOPs represents the number of floating-point operations required to infer 1 images. The highest scores for proprietary and open-source MLLMs are marked in bold. #Token refers to the token count for one image.

## C.3 RESULTS ANALYSIS

As shown in Table 10, LongLLaVA (single image) generally outperforms LLaVA-1.6-13B, despite both models having the same inference parameter size. This advantage is particularly notable in the MMMU benchmarks, highlighting LongLLaVA (single image)'s strengths in handling comprehensive knowledge-based questions. Although LongLLaVA (single image)'s performance is slightly lower compared to some recently emerged high-performing models, it still demonstrates the potential of hybrid architectures in multi-model scenarios. To ensure complete reproducibility of our results, we only focus on four representative public datasets. Additionally, we find that LongLLaVA tends to underperform relative to LongLLaVA (single image). Addressing this issue may require incorporating more single-image data during the Multi-image Instruction-tuning phase.

# D MULTIMODAL NEEDLE-IN-THE-AYSTACK EVALUATION

Using the V-NIAH evaluation framework proposed in LongVA (Zhang et al., 2024b), we conduct a needle-in-the-haystack test to evaluate the model's performance. As shown in Figure 9, LongLLaVA achieves nearly 100% retrieval accuracy on a set of 1200 images without requiring additional training.

---

[8]`gpt-4-vision-preview`
[9]`gemini-1.5-flash`
[10]`claude-3-opus-20240229`
[11]`https://huggingface.co/microsoft/Phi-3-vision-128k-instruct`

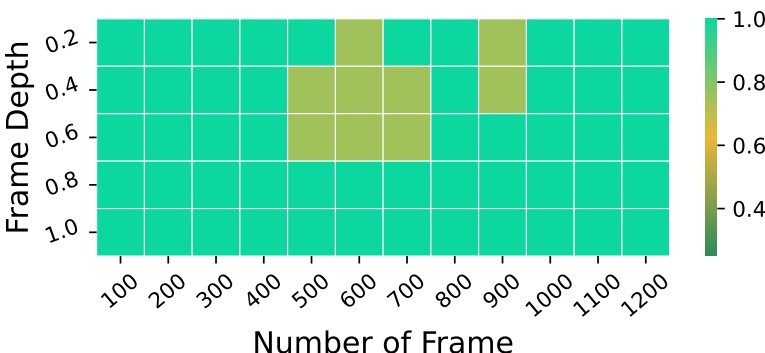

Figure 9: Video-NIAH (Zhang et al., 2024b) evaluated on one A800 80GB GPU.

# E    ADDITIONAL ABLATION STUDIES

## E.1    ABLATION OF TOKEN NUMBER PER IMAGE.

As demonstrated in Figure 10, setting 144 tokens per image effectively maintains performance while significantly reducing inference costs, particularly noticeable in the case of SEEDBench. Regarding **data construction**, after training on our single-image data, the model achieved a 1.5% accuracy improvement on SEEDBench and 12.3% on MileBench. Subsequent multi-image training led to a further 7.4% increase on MileBench, validating the dataset construction's effectiveness.

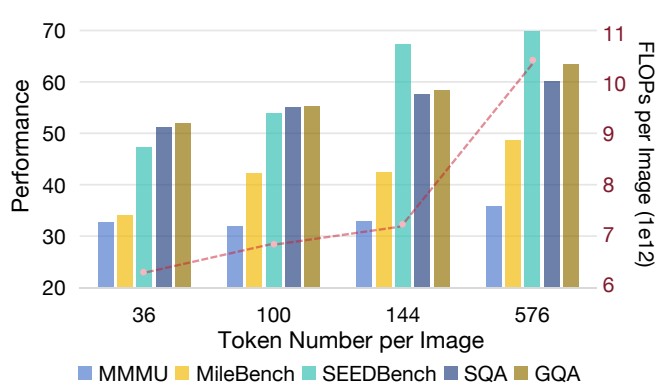

Figure 10: Performance across five datasets and inference costs with varying token numbers per image.

## E.2    ARCHITECTURE ABLATION STUDY ON 9B DENSE MODEL

To investigate whether the hybrid architecture impacts MLLM performance, we conducted experiments based on a 9B dense model. Ensuring alignment in the initial performance of the LLMs prior to MLLM adaptation is essential. Thus, we first compared the initial performance of the LLMs before adaptation. The results, presented in Table 11, demonstrate that the two models exhibit comparable performance.

Table 11: Initial Performance Comparison of LLMs

|            | MMLU | BBH  |
|------------|------|------|
| **Vicuna-13B** | **55.3** | **40.5** |
| **Jamba-9B**   | 54.3 | 38.4 |

Table 12: Performance Comparison of Different Architectures on Multimodal Benchmarks

| Model | GQA | MMMU | SQA$^I$ | SEED$^{v1}_{img}$ | Mile$^*_{avg}$ |
|-------|-----|------|---------|-------------------|----------------|
| **LLaVA-1.5-13B** | **63.3** | 34.4 | 71.6 | 68.2 | 27.6 |
| **Jamba-9B** (+LLaVA-1.5 Recipe) | 62.3 | **36.2** | **71.9** | **70.1** | **28.2** |
| **Difference** | -1.0 | +1.8 | +0.3 | +1.9 | +0.6 |

Subsequently, we conducted an additional ablation experiment by replacing the LLM base in LLaVA-1.5-13B with Jamba-9B (after pure-text instruction tuning) while following the LLaVA-1.5 training recipe. As shown in Table 12, given the comparable initial performance of the LLMs and

using the same training data combination, **the hybrid architecture achieves competitive results**. Furthermore, the hybrid architecture requires fewer FLOPs for inference.

E.3    REPLAY DATA ABLATION STUDY

To assess the impact of replay data, we conducted three experiments as part of the Replay Data Ablation Study.

Table 13: Comparison of Model Performance With and Without Replay Data

|  | MMLU | BBH | GQA | MMMU | SQA$^I$ | SEED$_{img}^{v1}$ | Mile$_{avg}^*$ |
|---|---|---|---|---|---|---|---|
| **LongLLaVA-9B** | **53.9** | **38.8** | **58.4** | **34.4** | **69.9** | **67.9** | 46.5 |
| **w/o Replay Data** | 52.3 | 36.2 | 57.5 | 31.2 | 53.5 | 64.3 | 46.8 |
| **Replace with Multi-Image** | 52.6 | 35.9 | 57.2 | 29.8 | 52.6 | 63.8 | **47.2** |

**Comparison With and Without Replay Data.**    We first conducted experiments comparing models trained with and without replay data. To isolate the effect of replay data from the impact of increased training data, we performed an ablation study by replacing replay data in the original training recipe with an equivalent amount of multi-image data. The results, presented in Table 13, demonstrate that **replay data is essential for preserving the model's original single-image understanding and text-following capabilities**.

Table 14: Impact of Text Replay Data Quantity

|  | MMLU | BBH |
|---|---|---|
| **LongLLaVA-9B (w/o Replay Data)** | 52.3 | 36.2 |
| **with 10K** | 52.9 | 37.3 |
| **with 20K** | 53.4 | 38.1 |
| **with 50K** | **53.9** | 38.8 |
| **with 100K** | 53.9 | **39.2** |

Table 15: Impact of Single-Image Replay Data Quantity

|  | GQA | MMMU | SQA$^I$ | SEED$_{img}^{v1}$ | Mile$_{avg}^*$ |
|---|---|---|---|---|---|
| **LongLLaVA-9B (w/o Replay Data)** | 57.5 | 31.2 | 53.5 | 64.3 | **46.8** |
| **with 50K** | 57.9 | 32.3 | 58.2 | 66.2 | 46.5 |
| **with 100K** | 57.9 | 33.5 | 62.7 | 67.1 | 46.5 |
| **with 200K** | 58.2 | 34.5 | 67.1 | 67.9 | **46.8** |
| **with 400K** | **58.5** | **35.2** | **73.2** | **68.2** | 46.4 |

**Replay Data Quantity Ablation.**    We also examined the impact of varying the quantity of replay data. For **text replay data**, the supplementary experiments reveal that adding text replay data enhances the model's text-following ability, although the improvement eventually saturates, as shown in Table 14. For **single-image replay data**, the results in Table 15 indicate that the model's single-image capability continues to improve with increased data volume and has not yet reached saturation. However, the improvement in multi-image tasks is limited.

