# OpenReview forum: "LongLLaVA: Scaling Multi-modal LLMs to 1000 Images Efficiently via a Hybrid Architecture"
_ICLR.cc/2025/Conference — Submitted to ICLR 2025_

### Official Review · Reviewer_Ut5k · 2024-11-02

**Soundness:** 3
**Presentation:** 3
**Contribution:** 2
**Rating:** 8
**Confidence:** 3

**Summary:**

The paper focuses on enhancing the long-context capabilities of Multi-modal Large Language Models (MLLMs), essential for tasks like video understanding, high-resolution image analysis, and multi-modal agents.

 The authors address key challenges, such as performance degradation with increased image inputs and high computational demands. They introduce a hybrid model architecture combining Mamba and Transformer blocks, optimize data construction to account for temporal and spatial dependencies, and use a progressive training strategy.

The resulting model, LongLLaVA (Long-Context Large Language and Vision Assistant), demonstrates competitive performance on multiple benchmarks, while also being memory-efficient and capable of processing nearly a thousand images on a single A100 80GB GPU, highlighting its potential for various applications.

**Strengths:**

1.The paper presents a comprehensive set of optimizations, including model architecture, data construction, and training strategy, showcasing a well-rounded and complete approach.

2.It features clear and well-structured illustrations alongside a compelling and coherent narrative.

3.The work addresses the urgent and impactful need for scaling up the number of images and handling long video generation, which is crucial for advancing the capabilities of MLLMs.

4.Extensive ablation studies are conducted, comparing the proposed model with both open-source and proprietary models across various benchmark datasets, including rigorous long-context tests like the Needle-In-A-Haystack evaluation.

**Weaknesses:**

1.The paper does not provide a comprehensive comparison of maximum throughput and latency profiles between the hybrid-structured model, pure Transformer-based models, and Mamba-based models, leaving a gap in understanding the performance trade-offs.

2.The extension of an existing, widely-used hybrid architecture to a new use case may reduce the perceived novelty of the work, as the contribution could be seen as incremental rather than groundbreaking.(see more details at questions)

**Questions:**

1. Could the authors provide a detailed comparison of system performance, specifically two graphs/tables maximum throughput and latency's relation on one/multiple GPUs, between the hybrid model, Transformer-based model, and Mamba-based model at a similar parameter size? This would give readers a clearer understanding of the efficiency gains and trade-offs associated with the proposed hybrid architecture.

2. I wonder if the authors could further emphasize the differences—such as model architecture, training methods, and other relevant aspects—between this work and Jamba, as well as other hybrid structured models, to better highlight the novelty of this paper.

---

> ### Author Response · Authors · 2024-11-25
> **Thanks for your kind reviews**
>
> Thank you for your insightful and helpful review. We undertake a series of systematic optimizations aimed at scaling Multi-modal LLMs to more than 1K Images efficiently. The model demonstrates impressive performance in potential application domains such as medicine and remote sensing. Below, we will provide detailed responses to each of your questions.
>
> **Q1: Absence of performance analysis among hybrid architecture models, pure Transformer models, and Mamba models**
>
> > We appreciate your constructive feedback. In the original version of the paper, Section 5.1 provided an analysis of the performance across various architectures, encapsulating factors such as prefill time, throughput, and memory usage.
> >
> > We have now incorporated an additional metric: the *maximum throughput* as suggested. This essentially measures the increase in batch size to the maximum permissible limit within memory constraints. Furthermore, in response to Reviewer RAbZ (the first reviewer)'s suggestion 1, we have also included an enhanced Mamba baseline in our study. Please refer to lines 384, 393-396 in the revised version.
> >
> > Below is a brief summary of the key findings from our extended analysis.
> >
> > | Model           | Arch.       | Active Param. | Prefill (s) | TP (tokens/s) | Mem.(GB) | Max TP (token/s) |
> > | --------------- | ----------- | ------------- | ----------- | ------------- | -------- | ---------------- |
> > | Cobra           | Mamba       | 3B            | 10.2        | 42.7          | 29.9     | 192.1            |
> > | Falcon-mamba[1] | Mamba       | 7B            | 24.3        | 32.4          | 48.8     | 90.7             |
> > | LLaVA-1.6       | Transformer | 13B           | 34.0        | 14.7          | 79.4     | 14.7             |
> > | LongLLaVA-9B    | Hybrid      | 9B            | 16.5        | 62.1          | 38.7     | 155.2            |
> > | LongLLaVA-A13B  | Hybrid      | 13B           | 25.5        | 37.6          | 79.1     | 37.6             |
> >
> > [1] Falcon Mamba: The First Competitive Attention-free 7B Language Model
>
> **Q2: Lack of emphasis on the novel contributions compared to existing hybrid architectures, potentially reducing the perceived novelty of the work.**
>
> > We appreciate the opportunity to clarify the novel contributions of our work. Our contributions can be viewed from two key perspectives: the impact on the **hybrid architecture community** and the **multimodal community**.
> >
> > - **Hybrid Architecture Community:** The hybrid architecture community has been seeking a compelling application to highlight its potential to the larger research audience. Our work leverages the long-context capabilities and efficient inference required by visual multimodal tasks, which align well with the strengths of hybrid architectures. We are the first to apply hybrid architectures to visual multimodal tasks, demonstrating the feasibility of modality expansion. This provides new insights and confidence for future optimization efforts within the community.
> >
> > - **Multimodal Community:** At the time of our work, there was no existing research enabling reasoning over more than 1,000 images using the LLaVA architecture, particularly with efficient real-time deployment. The community lacked guidance on several critical elements, such as designing training strategies for optimal performance, sourcing and processing training data, and establishing appropriate data protocols. Our work, alongside contemporaneous efforts, addresses these gaps. We have conducted one of the few systematic optimization studies focused on processing 1,000 images efficiently. Unlike other methods with inference times spanning tens of minutes to hours, our approach processes 1,000 images in under 3 minutes, facilitating practical deployment and enhancing the model's utility for the community.
> >
> > Section 6 of the paper outlines the application of LongLLaVA in domains such as medical and remote sensing imagery. The model achieves near state-of-the-art performance with only a few hours of training, and also excels in long-tail scenarios through a many-shot in-context learning (ICL) approach without fine-tuning. This highlights the robustness and efficiency of our approach as a foundational model, driving advancements in related fields and optimizing task-specific performance.
> >

---

> > ### Comment · Reviewer_Ut5k · 2024-12-02
> > **Response**
> >
> > Thanks for the response. I would like to keep the score.

---

> > > ### Author Response · Authors · 2024-12-02
> > >
> > > Thanks for your kind reviews.

---

### Official Review · Reviewer_UWnW · 2024-11-03

**Soundness:** 2
**Presentation:** 2
**Contribution:** 3
**Rating:** 5
**Confidence:** 3

**Summary:**

The work introduces LongLLaVA, a multimodal large language model that aims to solve challenges related to scaling up the number of in-context images in similar models as well as study design choices in the architecture, data, and training recipe of multimodal LMs. The challenges and motivations are outlined clearly and the authors choose to explore a hybrid architecture inspired by recent work around state space models, token compression through 2D pooling, and various data augmentation strategies. The authors evaluate the work on three multi-image benchmarks and show strong performance against open-source baselines and comparable performance to some closed models, despite being much more efficient to serve. The authors run an ablation study on the architecture choices and showcase additional interesting applications for this new model.

**Strengths:**

- The motivations are outlined very clearly and the way the authors chose to address the challenges presented makes sense
- Hybrid architecture efficiency analysis
- Ablation of architecture choices
- Multiple model scales presents some scaling analysis
- Many experiments, including additional applications to healthcare and science

**Weaknesses:**

- There is no ablation study showcasing the 3-stage finetuning vs a typical 1-stage finetuning step with all of the data across the three stages mixed in.

**Questions:**

- Are there any qualitative examples where this method performs better or worse than the baselines? Are there certain subtasks that this method is particularly well-suited for?
- As a result of pooling are there any tasks that suffer as a result? It could be helpful to include OCR or fine-grained image recognition tasks
- Does this new architecture show improved many-shot ICL performance (https://arxiv.org/abs/2404.11018) as opposed to finetuning?

---

> ### Author Response · Authors · 2024-11-25
> **Thanks for your kind reviews (1/N)**
>
> Thank you for your insightful and helpful review. We undertake a series of systematic optimizations aimed at scaling Multi-modal LLMs to more than 1K Images efficiently. The model demonstrates impressive performance in potential application domains such as medicine and remote sensing. Below, we will provide detailed responses to each of your questions.
>
> **Q1: Lack of an ablation study for the 3-stage versus 1-stage fine-tuning process**
>
> > - **Clarification of the Training Stages:**
> >   To provide a comprehensive understanding of the training stages, we summarize the divisions used in related works.
> >
> >   | Model     | Stage 1                           | Stage 2                                         | Stage 3                                   | Stage 4                    | Stage 5                                |
> >   | --------- | --------------------------------- | ----------------------------------------------- | ----------------------------------------- | -------------------------- | -------------------------------------- |
> >   | LLaVA     | Multimodal Alignment Pre-training | Visual Instruction Tuning                       |                                           |                            |                                        |
> >   | InternVL  | Contrastive Pre-training          | Generative Pre-training                         | Visual Instruction Tuning                 |                            |                                        |
> >   | Qwen-VL   | Multimodal Alignment Pre-training | Multi-task Pre-training                         | Visual Instruction Tuning                 |                            |                                        |
> >   | VILA      | Multimodal Alignment Pre-training | Multimodal Alignment Pre-training (interleaved) | Visual Instruction Tuning                 |                            |                                        |
> >   | LongVA    | Context Extension for LLMs        | Aligning Long LM with Short Vision Data         |                                           |                            |                                        |
> >   | LongVILA  | Multimodal Alignment Pre-training | Multimodal Alignment Pre-training (interleaved) | Visual Instruction-tuning (short context) | Context Extension for LLMs | Visual Instruction Tuning (long video) |
> >   | LongLLaVA | Multimodal Alignment Pre-training | Visual Instruction Tuning (single-image)        | Visual Instruction Tuning (multi-image)   |                            |                                        |
> >
> >   Although different works adopt varying stage divisions, they all follow the intuitive progression from simpler to more complex tasks. LongLLaVA adheres to this principle by training first on single images and then on multiple images, aligning with curriculum learning insights. While single-stage training offers a simpler protocol and continuous learning rates, multi-stage training allows for step-by-step verification of the model's capabilities, making the process more controlled.
> >
> > - **Supplementary Ablation Experiments:**
> >   In response to your suggestion, we conducted an ablation study to evaluate the necessity of separating training stages, using LongLLaVA-9B.
> >   The results, detailed in the accompanying table, show that multi-stage training achieves better performance on multi-image tasks while maintaining comparable results on single-image tasks. This finding aligns with insights from Section 3.3 of Cambrian-1[1], which also adopted a multi-stage training approach. The relevant results and discussion are presented in lines 365, 370-373 of the revised version of the paper.
> >
> >   | Stage                                      | GQA      | MMMU     | ScienceQA | SeedBench | Milebench | VideoMME |
> >   | ------------------------------------------ | -------- | -------- | --------- | --------- | --------- | -------- |
> >   | Stage1, 2, 3                               | **58.4** | **34.4** | 69.9      | 67.9      | **46.5**  | **43.7** |
> >   | Stage1, 2&3                                | 57.6     | 33.2     | **70.2**  | **68.4**  | 44.2      | 42.3     |
> >   | Stage1&2&3                                 | 56.9     | 32.8     | 67.2      | 66.9      | 42.2      | 40.1     |
> >   | & refers to the combination of the stages. |          |          |           |           |           |          |
> >
> >
> >
> > [1] Cambrian-1: A Fully Open, Vision-Centric Exploration of Multimodal LLMs

---

> ### Author Response · Authors · 2024-11-25
> **Thanks for your kind reviews (2/N)**
>
> **Q2: Lack of qualitative case studies and specificity on optimal subtasks for the proposed method.**
>
> > Thank you for this valuable feedback to enhance the paper's readability. We provide a detailed explanation below:
> >
> > - **Qualitative Examples:** Due to regulatory constraints, we included two examples in Section 6, which demonstrates the applications of LongLLaVA. Please refer to lines 498-509 of the revised version.
> >
> > - **Well-Suited Subtasks:** Among all models, LongLLaVA excels in multimodal long-context tasks, particularly in long-context retrieval tasks. As shown in Table 3, it significantly outperforms both proprietary and open-source models on retrieval-related tasks.
> >
>
> **Q3: Missing analysis on tasks affected by pooling.**
>
> > - **Performance Impact:** As indicated in Table 4, the model's performance on GQA, ScienceQA, SEEDBench, and MileBench degrades when using 2D Pooling (row 4) compared to the non-pooling baseline (row 2).
> >
> > - **Patching Strategy:** The degradation caused by pooling can be mitigated using our image patching strategy. For instance, we assessed the model's ability to identify small objects within large images using the V* Bench in Section 5.2. As Figure 5 illustrates, directly resizing large images into smaller ones for understanding limits the model due to the pooling strategy, making it challenging to locate small objects (Sub-image Count = 1). However, by slicing large images and treating them as multi-image tasks, we significantly improved performance by 18.9% (Sub-image Count = 97). For illustrative examples, please also refer to Figure 7 in Section 6.2 showing how slicing enhances the model’s ability to capture finer details.
> >
>
> **Q4: Question on whether this new architecture show improved many-shot ICL performance as opposed to finetuning.**
>
> > Thank you for this insightful suggestion, which prompted us to further explore the model's long-context capabilities.  In response, we expanded the evaluation of ICL capability in Section 5.1 of the original paper and detailed in Section 6.3 in the revised version as an application. Since LLMs fine-tuning can be costly and time-consuming, especially when data is scarce or frequent updates are needed. In contrast, Many-shot ICL allows models to use more task-specific examples during inference without retraining. We compared LongLLaVA-9B performance with different shot counts to fine-tuning on the same number of samples.
> >
> > |                | VL-ICL 1 Shot | VL-ICL 5 Shot | VL-ICL 10 Shot | VL-ICL 50 Shot | VL-ICL 100 Shot | VL-ICL 1000 Shot |
> > | -------------- | ------------- | ------------- | -------------- | -------------- | --------------- | ---------------- |
> > | LongLLaVA-9B   | **51.6**      | **60.2**      | **62.6**       | **66.5**       | **69.2**        | 70.4             |
> > | After Training | 50.1          | 53.1          | 53.5           | 60.2           | 68.9            | **71.3**         |
> >
> >  As shown in table above, LongLLaVA-9B outperforms ICL up to 100 shots. Beyond 1000 shots, the benefit of adding more shots diminishes, and fine-tuning becomes more effective. This suggests that ICL is preferable with fewer than 100 samples, while fine-tuning is more beneficial with around 1000 samples.

---

> ### Author Response · Authors · 2024-11-29
> **Follow-Up on Review and Feedback**
>
> Dear Reviewer **UWnW**,
>
> We hope this message finds you well.
>
> We have carefully addressed all your questions and concerns, including conducting additional experiments as requested, and have provided detailed responses in the rebuttal.
>
> As the rebuttal deadline is approaching, we would deeply appreciate it if you could share your updated thoughts based on the rebuttal and paper revision, or do not hesitate to let us know if you have additional questions, and we will respond promptly.
>
> Thank you again for your thoughtful review and your invaluable contributions to the quality of this paper.
>
> Kind regards,
>
> Paper 10001 Authors

---

> > ### Comment · Reviewer_UWnW · 2024-12-02
> >
> > Thank you for the thoughtful follow-up. I would like to keep my score.

---

> ### Author Response · Authors · 2024-12-02
>
> Do you have any thoughtful suggestions for rebuttal itself? If you have any other questions or need further clarification, please feel free to let us know. Your insights are highly valued. We would greatly appreciate it if you could express your concerns or questions clearly.

---

> ### Author Response · Authors · 2024-12-03
>
> Are there any concerns or issues that are causing you to remain weak rejection instead of giving positive one? If not, please responsibly reconsider your score as it is important to us. If you have any other questions or need further clarification, please feel free to let us know. Your insights are highly valued.
>
> From the perspective of community progress, weak rejection without any reason is not a responsible behavior in the community, which will result in both parties and the public being unable to benefit from communication.
>
> We would greatly appreciate it if you could express your concerns or questions clearly.

---

### Official Review · Reviewer_jB9q · 2024-11-04

**Soundness:** 3
**Presentation:** 2
**Contribution:** 2
**Rating:** 5
**Confidence:** 5

**Summary:**

This paper presents LongLLaVA, a MLLM designed for efficient long-video understanding. The proposed design applies a hybrid architecture inspired by Jamba that combines Mamba and Transformer layers, with 2D pooling applied for image token compression, and follows a progressive multi-stage training strategy. The paper conducts experiments in various benchmarks, including retrieval, counting, and video analysis tasks, while maintaining efficient resource use on an 80GB GPU for long video inference.

**Strengths:**

- The integration of Mamba and Transformer layers enables LongLLaVA to achieve quasi-linear computational complexity while supporting in-context learning. This design is well motivated to deal with long videos.
- Detailed experimental results, ablation studies, and diagnostic evaluations have been conducted on benchmarks like MileBench, VNBench, and Video-MME, showcasing that LongLLaVA has reasonable performances on multi-image and video tasks.
- LongLLaVA maintains a lower computational cost compared to full-attention counterparts, showing its potential in cost-effective deployment.

**Weaknesses:**

- The use of LongLLaVA-9B from Expert-0 in the Mamba (Jamba) MoE Layer appears unorthodox and lacks sufficient justification. The VLM's capabilities are heavily dependent on its underlying LLM, and it's unclear how well Jamba expert-0 performs as an LLM.
- The evaluation baselines are outdated and not sufficiently competitive. While LongVA-7B is mentioned, it's not directly compared against, and several newer 7B VLMs with superior performance are excluded. Although fair comparisons are challenging due to varying training data mixtures and model backbones, the authors should provide a more comprehensive overview of existing VLMs of similar size, including LongVA-7B at the least.
- More rigorous experimental design is needed to demonstrate that the hybrid architecture doesn't compromise VLM performance. While Tables 4 and 5 show LongLLaVA-9B outperforming LLaVA-1.5-13B and LLaVA-1.6-13B, this comparison is skewed since Jamba already outperforms LLaMA2-13B (and likely Vicuna-1.5) in LLM benchmarks [1]. Additionally, the VLM training data mixtures differ. To validate that the hybrid architecture matches traditional full-attention LLM performance as a VLM backbone, LLMs models in those two architectures with similar capabilities (e.g., Zamba-2-7B [2] and LLaMA3.1-8B) should be trained using the LongLLaVA training recipe.
- The training strategy lacks sufficient ablation studies, particularly regarding the impact of removing replay data.
- Tables require additional context. Tables 2 and 3 should include model sizes for open-source baselines. While PFLOPs indicates model efficiency, model sizes are crucial for understanding expected performance.

[1] Lieber et al. Jamba: A Hybrid Transformer-Mamba Language Model.
[2] https://huggingface.co/Zyphra/Zamba2-7B

**Questions:**

- What does 1D Pooling in table 4 specifically refer to?
- What does LLaVA-1.5-13B + Jamba in table 4 mean? Is it just replacing LLaVA-1.5-13B’s LLM with Jamba? Are those both trained with the LongLLaVA recipe?

---

> ### Author Response · Authors · 2024-11-25
> **Thanks for your kind reviews (1/N)**
>
> Thank you for your insightful and helpful review. We undertake a series of systematic optimizations aimed at scaling Multi-modal LLMs to more than 1K Images efficiently. The model demonstrates impressive performance in potential application domains such as medicine and remote sensing. Below, we will provide detailed responses to each of your questions.
>
> **Q1: The selection of Expert-0 from the MoE layers for LongLLaVA-9B lacks sufficient justification.**
>
> > Thank you for highlighting this concern. We chose Expert-0 based on preliminary experiments indicating minimal differences in model performance (measured by MMLU [1] and BBH [2] accuracy) after Pure-text Instruction Tuning, regardless of whether numerical averaging, spherical averaging, or random expert selection was employed. For simplicity, we opted to use Expert-0. As this experiment was not central to the main focus of the paper, detailed descriptions were initially omitted. The relevant experimental results are shown below, and in Appendix A and line 215 of the revised version.
> >
> > | **Downcycling Strategy** | **Arithmetic Mean** | **Spherical Mean** | **Expert-0** | **Expert-5** | **Expert-12** | **Expert-15** |
> > | ------------------------ | ------------------- | ------------------ | ------------ | ------------ | ------------- | ------------- |
> > | MMLU                     | 52.7                | **53.2**           | **53.2**     | 51.9         | 52.6          | 52.2          |
> > | MMLU Aft. Train          | 53.8                | **54.3**           | **54.3**     | 53.3         | 53.8          | 53.3          |
> > | BBH                      | 36.7                | 36.7               | 37.2         | 36.7         | **37.4**      | 36.3          |
> > | BBH Aft. Train           | 37.8                | 37.9               | 38.4         | **38.9**     | **38.9**      | 37.9          |
> >
> > [1] Measuring Massive Multitask Language Understanding
> >
> > [2] Challenging BIG-Bench Tasks and Whether Chain-of-Thought Can Solve Them
>
>
>
> **Q2: Lack of LongVA Baseline Results and Discussion About Performance**
>
> > - **Lack of LongVA Baseline:**
> >
> >   We apologize for any inconvenience this may have caused. As noted in the LongVA paper, the LongVA model was trained on synthetic multi-image datasets where images are divided into sub-images, leading to its initial exclusion from our performance comparisons. Following your suggestion, we have now included LongVA in our evaluation to provide a clearer view of its performance relative to other models. Key results are presented in the table below.
> >
> > - **Discussion About Performance:**
> >
> >   As you mentioned, model performance is significantly influenced by factors such as training time, training data, inference FLOPs, and inference precision. LongLLaVA-9B uses the same training data and INT8 inference precision as LongLLaVA-A13B to ensure alignment. The multimodal adaptation training time for LongLLaVA-9B is 46h × 8 × A800-80G, which is 54.8% of LongVA's (84h × 8 × A100-80G) and significantly lower than LongVILA's. Additionally, the FLOPs required for inference are only 3.8% of those needed for the other two models.
> >
> >   However, as reviewer RAbZ pointed out, this presentation format limits the scope and accessibility of the results for the community. To address this, we have taken two steps: first, we provide the models' FP-16 precision inference results for reference. Second, we introduce a variant, LongLLaVA-9B-MoreData, trained on more data (82h × 8 × A800-80G). Key results are summarized in the following table, demonstrating that the LongLLaVA series achieves comparable performance with high efficiency.  Please refer to Table 2 for details.
> >
> >   | Model                       | Precision | PFLOPs | VideoMME-Short | VideoMME-Medium | VideoMME-Long | VideoMME-Avg. |
> >   | --------------------------- | --------- | ------ | -------------- | --------------- | ------------- | ------------- |
> >   | LongVILA                    | FP16      | 3.90   | 61.8           | 49.7            | 39.7          | 50.5          |
> >   | **LongVA**                  | FP16      | 4.90   | 61.1           | 50.4            | 46.2          | 52.6          |
> >   | LongLLaVA-9B                | INT8      | 0.15   | 52.4           | 42.2            | 36.4          | 43.7          |
> >   | LongLLaVA-A13B              | INT8      | 0.22   | 60.9           | 49.7            | 44.1          | 51.6          |
> >   | **LongLLaVA-9B (MoreData)** | FP16      | 0.15   | 58.4           | 48.3            | 41.7          | 49.5          |
> >   | **LongLLaVA-A13B**          | FP16      | 0.22   | **62.9**       | **52.2**        | **46.4**      | **53.8**      |
> >

---

> ### Author Response · Authors · 2024-11-25
> **Thanks for your kind reviews (2/N)**
>
> **Q3: How can we ensure the hybrid architecture maintains VLM performance given the skewed comparisons due to differences in baseline performance and VLM training data?**
>
> > - **Clarification on LLaVA-1.5-13B + Jamba in Table 4:** This refers to replacing LLaVA-1.5-13B's LLM (Vicuna-1.5) with Jamba (after Pure-text Instruction Tuning), with both trained using the LLaVA-1.5 recipe.
> >
> > - We understand your concerns, and in our ablation study, we analyzed the impact of the LLM model **using the same training recipe**. However, as you noted, aligning the initial performance of the LLMs prior to VLM adaptation is crucial. Therefore, we conducted an additional ablation experiment by **replacing the LLM base in LLaVA-1.5-13B with Jamba-9B (after pure-text instruction tuning)** while following the LLaVA-1.5 training recipe. The initial performance comparison of the large models before VLM adaptation is presented below, showing that **the two models exhibit comparable performance**.
> >
> >   |                | **MMLU** | **BBH**  |
> >   | -------------- | -------- | -------- |
> >   | **Vicuna 13B** | **55.3** | **40.5** |
> >   | **Jamba-9B**   | 54.3     | 38.4     |
> >
> > - The key experimental results of the related ablation are summarized below. As shown in the table, given the comparable initial performance of the LLMs and using the same training data combination, **the hybrid architecture achieves competitive results**. Additionally, the hybrid architecture requires fewer FLOPs for inference.
> >
> >   | Model                                                    | **GQA**  | **MMMU** | **ScienceQA** | **SeedBench** | **MileBench** |
> >   | -------------------------------------------------------- | -------- | -------- | ------------- | ------------- | ------------- |
> >   | LLaVA-1.5-13B                                            | **63.3** | 34.4     | 71.6          | 68.2          | 27.6          |
> >   | Jamba-9B (same data and training strategy with LLaVA1.5) | 62.3     | **36.2** | **71.9**      | **70.1**      | **28.2**      |
> >   | **Difference**                                           | -1.0     | +1.8     | +0.3          | +1.9          | +0.6          |
> >
> > The relevant experimental details and discussion can be found in lines 372-373 of the revised version and Appendix E.2.
> >

---

> ### Author Response · Authors · 2024-11-25
> **Thanks for your kind reviews (3/N)**
>
> **Q4: A lack of ablation studies on the training recipes, particularly regarding the Replay data**
>
> > - **On Merging or Separating Training Stages:**
> >   Detailed experimental results on whether the various stages should be merged or separated are provided in our response to Reviewer RAbZ (the first reviewer)'s Q3. To enhance your reading experience, we will focus here on the discussion of Replay data.
> >
> > - **Impact of Replay Data:**
> >
> >   - **Comparison With and Without Replay Data:**
> >     As requested, we conducted experiments comparing models trained with and without Replay Data. To eliminate the influence of increased training data introduced by Replay Data, we also conducted an ablation study replacing Replay Data in the original training recipe with equivalent multi-image data. The results, presented in the table below, demonstrate that **Replay Data is crucial for maintaining the model's original single-image understanding and text-following capabilities**.
> >
> >     |                                 | MMLU     | BBH      | GQA      | MMMU     | ScienceQA | SeedBench | MileBench |
> >     | ------------------------------- | -------- | -------- | -------- | -------- | --------- | --------- | --------- |
> >     | LongLLaVA-9B                    | **53.9** | **38.8** | **58.4** | **34.4** | **69.9**  | **67.9**  | 46.5      |
> >     | w/o Replay Data                 | 52.3     | 36.2     | 57.5     | 31.2     | 53.5      | 64.3      | 46.8      |
> >     | replace Replay with Multi-Image | 52.6     | 35.9     | 57.2     | 29.8     | 52.6      | 63.8      | **47.2**  |
> >
> >   - **Replay Data Quantity Ablation:**
> >     Based on your suggestion, we also explored the impact of the quantity of Replay Data.
> >
> >       - **Text:** The supplementary experiment examining the impact of data quantity shows that **adding text replay data enhances the model's text-following ability, although the improvement eventually saturates**.
> >
> > 	  |                                | MMLU     | BBH      |
> > 	  | ------------------------------ | -------- | -------- |
> > 	  | LongLLaVA-9B (w/o Replay Data) | 52.3     | 36.2     |
> > 	  | with 10K Text Replay Data      | 52.9     | 37.3     |
> > 	  | with 20K Text Replay Data      | 53.4     | 38.1     |
> > 	  | with 50K Text Replay Data      | **53.9** | 38.8     |
> > 	  | with 100K Text Replay Data     | 53.9     | **39.2** |
> >
> > 	  - **Single-Image:** The experiment indicates that the model's single-image capability continues to improve with increased data volume and **has not yet reached saturation**. However, the improvement in multi-image tasks is limited.
> >
> > 	    |                                    | GQA      | MMMU     | ScienceQA | SeedBench | MileBench |
> > 	    | ---------------------------------- | -------- | -------- | --------- | --------- | --------- |
> > 	    | LongLLAVA-9B (w/o Replay Data)     | 57.5     | 31.2     | 53.5      | 64.3      | **46.8**  |
> > 	    | with 50K Single-Image Replay Data  | 57.9     | 32.3     | 58.2      | 66.2      | 46.5      |
> > 	    | with 100K Single-Image Replay Data | 57.9     | 33.5     | 62.7      | 67.1      | 46.5      |
> > 	    | with 200K Single-Image Replay Data | 58.2     | 34.5     | 67.1      | 67.9      | **46.8**  |
> > 	    | with 400K Single-Image Replay Data | **58.5** | **35.2** | **73.2**  | **68.2**  | 46.4      |
> >
> > The relevant experimental details and discussion can be found in lines 372-373 of the revised version and Appendix E.3.
> >
>
>
>
>
>
> **Q5: What does 1D Pooling in Table 4 specifically refer to?**
>
> > 1D pooling involves flattening the image pixels into a sequence and applying pooling operations directly to four consecutive pixels, unlike 2D pooling, which preserves the relative positional information of the image space by pooling in units of 2×2 squares.
> >
> > Here are examples of 1D and 2D pooling:
> >
> > - **1D Pooling Example:**
> >
> >   For the following 2D array, 1D pooling applies a max pooling operation on every 4 consecutive pixels in each row:
> >
> >   ```
> >   Original Array:
> >   [[ 1,  2,  3,  4]
> >    [ 5,  6,  7,  8]
> >    [ 9, 10, 11, 12]
> >    [13, 14, 15, 16]]
> >   ```
> >
> >   **1D Pooling Result (Max pooling across each row with 4 values):**
> >
> >   ```
> >   [[ 4,  8, 12, 16]]
> >   ```
> >
> > - **2D Pooling Example:**
> >
> >   For the same array, 2D pooling applies a max pooling operation on 2x2 blocks:
> >
> >   ```
> >   Original Array:
> >   [[ 1,  2,  3,  4]
> >    [ 5,  6,  7,  8]
> >    [ 9, 10, 11, 12]
> >    [13, 14, 15, 16]]
> >   ```
> >
> >   **2D Pooling Result (Max pooling with 2x2 blocks):**
> >
> >   ```
> >   [[ 6,  8]
> >    [14, 16]]
> >   ```
> >
> > In summary, 1D pooling pools along rows of the flattened sequence, while 2D pooling operates on 2x2 blocks, maintaining spatial information.
> >

---

> ### Author Response · Authors · 2024-11-29
> **Follow-Up on Review and Feedback**
>
> Dear Reviewer **jB9q**,
>
> We hope this message finds you well.
>
> We have carefully addressed all your questions and concerns, including conducting additional experiments as requested, and have provided detailed responses in the rebuttal.
>
> As the rebuttal deadline is approaching, we would deeply appreciate it if you could share your updated thoughts based on the rebuttal and paper revision, or do not hesitate to let us know if you have additional questions, and we will respond promptly.
>
> Thank you again for your thoughtful review and your invaluable contributions to the quality of this paper.
>
> Kind regards,
>
> Paper 10001 Authors

---

> ### Author Response · Authors · 2024-12-02
>
> Dear Reviewer jB9q,
>
> We hope this message finds you well.
>
> We have carefully addressed all your questions and concerns, including conducting additional experiments as requested, and have provided detailed responses in the rebuttal.
>
> As the rebuttal deadline is approaching, we would deeply appreciate it if you could share your updated thoughts based on the rebuttal and paper revision, or do not hesitate to let us know if you have additional questions, and we will respond promptly.
>
> Thank you again for your thoughtful review and your invaluable contributions to the quality of this paper.
>
> Kind regards,
>
> Paper 10001 Authors

---

> ### Author Response · Authors · 2024-12-03
>
> Dear Reviewer jB9q,
>
> We hope this message finds you well.
>
> We have carefully addressed all your questions and concerns, including conducting additional experiments as requested, and have provided detailed responses in the rebuttal.
>
> As the rebuttal deadline is approaching, we would deeply appreciate it if you could share your updated thoughts based on the rebuttal and paper revision, or do not hesitate to let us know if you have additional questions, and we will respond promptly.
>
> Thank you again for your thoughtful review and your invaluable contributions to the quality of this paper.
>
> Kind regards,
>
> Paper 10001 Authors

---

### Official Review · Reviewer_RAbZ · 2024-11-06

**Soundness:** 2
**Presentation:** 3
**Contribution:** 2
**Rating:** 5
**Confidence:** 5

**Summary:**

The paper presents LongLLaVA, a novel solution to enhance the long-context capabilities. Architecture-wise, it combines Mamba (pseudo attention) and token compression.  It can process >1000 images on a single A100 80GB GPU. Experimental results show good performance in multi-modal long-context understanding tasks, surpassing many models in MileBench and VNBench. It also demonstrates strong application potential in healthcare and science domains

**Strengths:**

LongLLaVA can handle up to 1173 images on a single 80GB GPU, showing excellent processing power of handling more images, enabling more spatial and temporal information.

The proposed efficient hybrid architecture improving throughput and reducing memory usage while maintaining good performance in both ICL and VLM benchmarks.

The enhanced data construction and progressive training strategy guide the model to distinguish temporal and spatial dependencies among images.

**Weaknesses:**

My main concern for the paper is that the motivation is not clearly justified. LongLLaVa aims to enable more frames (tokens) for Vision Language Models (VLM) and thus explores a Mamba-based architecture. However, the reason for choosing a hybrid architecture is confusing. Lines 127-128 mention the Mamba model's in-context learning (ICL) capability as indispensable. Is there any evidence or literature to support that this is a weakness of the Mamba architecture itself rather than a result of training data? Additionally, Cobra in Table 5 only has results with a 3B size model, while other models are 9B–13B. This is not a fair comparison and doesn't convincingly justify the choice of a hybrid architecture.

While supporting over 1,000 images on an A100 GPU looks promising, the performance is not satisfying. LongLLaVa-9B-MoE (with actual parameters of 53B) shows a significant performance degradation compared to LongVILA (VideoMME Average drops from 50.5 to 43.7). The 13B MoE model is also not comparable with LongVA (released in July and cited in the paper but not compared). Furthermore, what is the performance of LongLLaVa on regular VLM benchmarks like DocVQA, ChartQA, and MMMU? The results in Table 4 and Figure 5 are from different models than those in Tables 4 and 5, and the comparison is incomplete.

Furthermore, the design choices are not well-ablated. For example, why does the training pipeline (Figure 4) train the model sequentially? Many previous VLM works like LongVA, VILA, and InternVL mix most data in one stage instead of feeding it sequentially. Another point is the architecture design: why blend transformer and Mamba in a 7:1 ratio and only apply MoE in Mamba? Do other blending ratios hurt the performance or lead to less memory saving? Could you provide ablations and share the thoughts behind the design choices?

**Questions:**

Needle in a haystack benchmark missing? What is retrival rate for LongLLaVa

How the model (in Figure 2) is initialized for later VLM training?

L123-124: ring-attention and sequence parallel are actual same technique with different naming.

L125: why sp or ring-atten introduce extra overhead? What is the comparison baseline here?

L127: Mama model … ICL capability … indispensable. Any evidence to support this is weakness from mamba arch itself rather than training data? Does this weakness exists in VLM or both LLM and VLM?

The paper titled LongLlava, seems to be like an extension of llava series work. But the architecture and training scheme have been both changed drastically.

---

> ### Author Response · Authors · 2024-11-25
> **Thanks for your kind reviews (1/N)**
>
> Thank you for your insightful and helpful review. We undertake a series of systematic optimizations aimed at scaling Multi-modal LLMs to more than 1K Images efficiently. The model demonstrates impressive performance in potential application domains such as medicine and remote sensing. Below, we will provide detailed responses to each of your questions.
>
> **Q1: The lack of evidence or literature supports that ICL is a weakness within the Mamba architecture. And the comparison in Table 5 about this is not fair.**
>
> > We address this question from two perspectives: literature support and additional experimental evidence.
> >
> > - **Literature Support on the Weakness of Mamba Architecture for ICL Tasks:**
> >
> >   Researches indicate that **Mamba architectures perform poorly in some ICL tasks**, as noted in [1] and [2]. Specifically, [1] highlights Mamba's limitations in tasks involving non-standard retrieval functionality and introduces hybrid architectures to address these issues. Study [2] suggests that ICL capabilities are linked to specific attention heads that emerge during training.
> >
> >   We appreciate your inclusion of the **training data dimension** in the ICL discussion. As mentioned, [1] highlights that while Mamba can learn to replicate simple ICL tasks through explicit training, this approach significantly constrains the full potential of the model’s parameters and training data.
> >
> > - **Literature Support for Hybrid Architectures:**
> >
> >   Interestingly, the authors of the Mamba series have also embraced hybrid architectures, highlighting their potential and performance advantages through comparative experiments (see Section 9.2.3 of Mamba-2 [3]). In a recent paper [4], they further support the distillation and acceleration of hybrid architectures, validating their performance advantages through extensive experiments.
> >
> > - **Additional Experiments:**
> >
> >   We acknowledge concerns regarding the fairness of comparing different architectures in Section 5.1. To address this, we compared our model against the largest available multimodal Mamba architecture at the time. Recently, thanks to the open-source community, we found a 7.3B Mamba language model [5] and trained and evaluated it using the same settings as our model. Due to the difficulty in flexibly adjusting the parameter count of MLLMs, we cannot fully align the number of parameters.
> >
> >   | Model        | Arch.       | Active Param. | VL-ICL 1 Shot | VL-ICL 2 Shot | VL-ICL 4 shot | VL-ICL 5-shot |
> >   | ------------ | ----------- | ------------- | ------------- | ------------- | ------------- | ------------- |
> >   | Falcon-mamba | Mamba       | 7B            | 49.0          | 51.9          | 52.4          | 53.2          |
> >   | LongLLaVA-9B | Hybrid      | 9B            | 51.6          | 57.8          | 58.4          | 60.2          |
> >   | LLaVA-1.6    | Transformer | 13B           | 50.0          | 52.3          | 54.6          | 58.9          |
> >
> >   The key results demonstrate that, as the shot number increased, the hybrid architecture model can effectively utilize the shots compared to Mamba architecture model, especially under the 5-shot setting.
> >
> >
> >
> > The above discussion and experimental results have been incorporated into the revised version of the paper, presented in a manner that aligns with the overall structure of the article. For further details, please refer to lines 121-137, 386, and 393-396 of the revised version.
> >
> > &nbsp;
> >
> > [1] Can mamba learn how to learn? a comparative study on in-context learning task
> >
> > [2] In-context Learning and Induction Heads
> >
> > [3] Transformers are SSMs: Generalized Models and Efficient Algorithms Through Structured State Space Duality
> >
> > [4] The Mamba in the Llama: Distilling and Accelerating Hybrid Models
> >
> > [5] Falcon Mamba: The First Competitive Attention-free 7B Language Model

---

> ### Author Response · Authors · 2024-11-25
> **Thanks for your kind reviews (2/N)**
>
> **Q2:The model performance is unsatisfactory, particularly for LongLLaVA-9B. Additionally, the baseline and evaluation are neither comprehensive nor adequate.**
>
> > - **Clarification:** LongLLaVA-9B is a dense model, not an MoE model.
> >
> > We apologize for any confusion caused by unclear writing in the original paper. To clarify, LongLLaVA-9B is a dense model, as indicated in lines 213–214 of the original version and lines 195-196 of the revised version.
> >
> > - **Comprehensive Evaluation:**
> >
> > In the original paper, we included **seven** standard VLM benchmarks, as detailed in Appendix B and discussed in lines 298–299. In the revised version, we have improved the presentation for better readability, with the related details now included in Appendix C and discussed in lines 303. Additionally, following your suggestion, we have added DocVQA and ChartQA to the existing benchmarks and enhanced the presentation logic to avoid further confusion. These updates are covered in lines 1041-1052 of the revised version.
> >
> >
> > - **Addition of LongVA Baseline:**
> >
> >   - **Performance Concerns:**
> >
> >     - LongLLaVA-9B uses the same training data and INT8 inference precision as LongLLaVA-A13B. The model's performance is influenced by training time, data, inference FLOPs, and inference precision. The multimodal adaptation training time for LongLLaVA-9B is 46 hours × 8 × A800-80G, which is 54.8% of LongVA's (84 hours × 8 × A100-80G) and significantly lower than LongVILA's. Additionally, the FLOPs required for inference are only 3.8% of those needed for the other two models.
> >
> >       To address presentation limitations, we now provide the models' FP-16 precision inference results for reference. Additionally, we introduce LongLLaVA-9B-MoreData, trained on the dataset for around two epochs (82 hours × 8 × A800-80G). The key results, summarized in the table below, demonstrate that the LongLLaVA series achieves comparable performance with high efficiency. Please refer to Table 2 for details.
> >
> >       | Model                        | Precision | PFLOPs | VideoMME-Short | VideoMME-Medium | VideoMME-Long | VideoMME-Avg. |
> >       | ---------------------------- | --------- | ------ | -------------- | --------------- | ------------- | ------------- |
> >       | LongVILA                     | FP16      | 3.90   | 61.8           | 49.7            | 39.7          | 50.5          |
> >       | **LongVA**                   | FP16      | 4.90   | 61.1           | 50.4            | 46.2          | 52.6          |
> >       | LongLLaVA-9B                 | INT8      | 0.15   | 52.4           | 42.2            | 36.4          | 43.7          |
> >       | LongLLaVA-A13B               | INT8      | 0.22   | 60.9           | 49.7            | 44.1          | 51.6          |
> >       | **LongLLaVA-9B (More Data)** | FP16      | 0.15   | 58.4           | 48.3            | 41.7          | 49.5          |
> >       | **LongLLaVA-A13B**           | FP16      | 0.22   | **62.9**       | **52.2**        | **46.4**      | **53.8**      |
> >
> >   - **Lack of LongVA Baseline:** We apologize for any inconvenience caused by the omission. The LongVA model was initially excluded from the performance comparison because it was not trained on real multi-image datasets, as noted in the LongVA paper. However, following your suggestion, we have now included LongVA in the evaluation to provide a clearer view of the progress in community model performance. Key results are presented in the table above.
> >

---

> ### Author Response · Authors · 2024-11-25
> **Thanks for your kind reviews (3/N)**
>
> **Q3:A lack of ablation studies on the multi-stage training recipes and insufficient clarification of the motivation behind model architecture design details.**
>
> > - **Motivation for Architecture Design Details:**
> >
> > In the hybrid architecture study by the authors of Mamba [1], various hybrid ratios (1:0, 7:1, 3:1, 1:1) were evaluated. The most significant benefit was observed in the transition from 1:0 to 7:1, with diminishing returns beyond that point. To optimize efficiency, we selected the 7:1 hybrid ratio. This decision aligns with findings in Section 6.1 of [2], where the performance difference between 7:1 and 3:1 was negligible, despite increased inference FLOPs. While we acknowledge the trade-offs involved with MoE, future investigations will explore these aspects further.
> >
> > - **Ablation Studies on Multi-stage Training Recipes:**
> >
> >   - **Clarification on "Multi-stage" Training:**
> >
> >     We summarize the training stage divisions presented in related works, which generally progress from simpler to more complex tasks.
> >
> > 	  | Model     | Stage 1                           | Stage 2                                         | Stage 3                                   | Stage 4                    | Stage 5                                |
> > 	  | - | - | - | - | - | - |
> > 	  | LLaVA     | Multimodal Alignment Pre-training | Visual Instruction Tuning                       |                                           |                            |                                        |
> > 	  | InternVL  | Contrastive Pre-training          | Generative Pre-training                         | Visual Instruction Tuning                 |                            |                                        |
> > 	  | Qwen-VL   | Multimodal Alignment Pre-training | Multi-task Pre-training                         | Visual Instruction Tuning                 |                            |                                        |
> > 	  | VILA      | Multimodal Alignment Pre-training | Multimodal Alignment Pre-training (interleaved) | Visual Instruction Tuning                 |                            |                                        |
> > 	  | LongVA    | Context Extension for LLMs        | Aligning Long LM with Short Vision Data         |                                           |                            |                                        |
> > 	  | LongVILA  | Multimodal Alignment Pre-training | Multimodal Alignment Pre-training (interleaved) | Visual Instruction-tuning (short context) | Context Extension for LLMs | Visual Instruction Tuning (long video) |
> > 	  | LongLLaVA | Multimodal Alignment Pre-training | Visual Instruction Tuning (single-image)        | Visual Instruction Tuning (multi-image)   |                            |                                        |
> >
> >       Processing single images serves as the foundation for handling multiple images. LongLLaVA adheres to this curriculum-learning insight by first training on single images before advancing to multiple images. While single-stage training offers a simpler protocol and continuous learning rates, multi-stage training allows for step-by-step verification of the model's capabilities, making the process more controllable. As suggested, we recognize the need for an ablation study to provide further clarity.
> >
> >   - **Supplementary Ablation Experiments:**
> >
> >     - We conducted an ablation experiment using LongLLaVA-9B to assess the necessity of separating training stages. The results, detailed in the accompanying table, indicate that **multi-stage training achieves better performance on multi-image tasks while maintaining comparable results on single-image tasks**. This finding aligns with insights from Section 3.3 of Cambrian-1[3], which also employed a multi-stage training approach. The relevant results and discussion are presented in lines 365, 370-373 of the revised version of the paper.
> >
> > 	    | Stage                                      | GQA      | MMMU     | ScienceQA | SeedBench | Milebench | VideoMME |
> > 	    | ------------------------------------------ | -------- | -------- | --------- | --------- | --------- | -------- |
> > 	    | Stage1, 2, 3                               | **58.4** | **34.4** | 69.9      | 67.9      | **46.5**  | **43.7** |
> > 	    | Stage1, 2&3                                | 57.6     | 33.2     | **70.2**  | **68.4**  | 44.2      | 42.3     |
> > 	    | Stage1&2&3                                 | 56.9     | 32.8     | 67.2      | 66.9      | 42.2      | 40.1     |
> > 	    | & refers to the combination of the stages. |          |          |           |           |           |          |
> >
> > [1] The Mamba in the Llama: Distilling and Accelerating Hybrid Models
> >
> > [2] Jamba: A Hybrid Transformer-Mamba Language Model
> >
> > [3] Cambrian-1: A Fully Open, Vision-Centric Exploration of Multimodal LLMs

---

> ### Author Response · Authors · 2024-11-25
> **Thanks for your kind reviews (4/N)**
>
> **Q4: Other Issues and Factual Clarifications**
>
> > - **Lack of Needle in a Haystack Benchmark:**
> >   The relevant results are provided in Appendix C of the original paper and referenced in lines 372–373. In the revised version, we increased the size of the corresponding image (1080 lines) and referenced in lines 348-350 of the text.
> >
> > - **Model Initialization for VLM Training (Figure 2):**
> >   The model is initialized following pure-text instruction tuning.
> >
> > - **Ring-Attention and Sequence Parallelism:**
> >   Ring-attention is a type of sequence parallelism method, with implementations similar to DeepSpeed Ulysses [1]. We will clarify this distinction in the revised paper.
> >
> > - **Extra Overhead from Ring-Attention or Sequence Parallelism:**
> >   Both sequence parallel methods introduce additional communication overhead.
> >
> > - **About the Model Name "LongLLaVA":**
> >   LongLLaVA (Long-Context Large Language and Vision Assistant) is a data-efficient multimodal adaptation approach, renowned for its multi-stage training methodology. We utilize efficient acceleration techniques to adapt it to long-context scenarios, and we express our respect and tribute to this outstanding work.
> >
> > [1] DeepSpeed Ulysses: System Optimizations for Enabling Training of Extreme Long Sequence Transformer Models

---

> ### Author Response · Authors · 2024-11-29
> **Follow-Up on Review and Feedback**
>
> Dear Reviewer **RAbZ**,
>
> We hope this message finds you well.
>
> We have carefully addressed all your questions and concerns, including conducting additional experiments as requested, and have provided detailed responses in the rebuttal.
>
> As the rebuttal deadline is approaching, we would deeply appreciate it if you could share your updated thoughts based on the rebuttal and paper revision, or do not hesitate to let us know if you have additional questions, and we will respond promptly.
>
> Thank you again for your thoughtful review and your invaluable contributions to the quality of this paper.
>
> Kind regards,
>
> Paper 10001 Authors

---

> ### Author Response · Authors · 2024-12-02
>
> Dear Reviewer RAbZ,
>
> We hope this message finds you well.
>
> We have carefully addressed all your questions and concerns, including conducting additional experiments as requested, and have provided detailed responses in the rebuttal.
>
> As the rebuttal deadline is approaching, we would deeply appreciate it if you could share your updated thoughts based on the rebuttal and paper revision, or do not hesitate to let us know if you have additional questions, and we will respond promptly.
>
> Thank you again for your thoughtful review and your invaluable contributions to the quality of this paper.
>
> Kind regards,
>
> Paper 10001 Authors

---

> ### Author Response · Authors · 2024-12-03
>
> Dear Reviewer RAbZ,
>
> We hope this message finds you well.
>
> We have carefully addressed all your questions and concerns, including conducting additional experiments as requested, and have provided detailed responses in the rebuttal.
>
> As the rebuttal deadline is approaching, we would deeply appreciate it if you could share your updated thoughts based on the rebuttal and paper revision, or do not hesitate to let us know if you have additional questions, and we will respond promptly.
>
> Thank you again for your thoughtful review and your invaluable contributions to the quality of this paper.
>
> Kind regards,
>
> Paper 10001 Authors

---

### Author Response · Authors · 2024-11-27
**General Response**

Dear Reviewers and ACs,

We sincerely thank all the reviewers and ACs for your diligent efforts and high-quality reviews. We appreciate the reviewers for recognizing the strengths of LongLLaVA, particularly its well-motivated hybrid architecture (Reviewers `RAbZ`, `UWnW`), efficient design (Reviewers `RAbZ`, `jB9q`) and strong performance (Reviewers `jB9q`, `UWnW`) with impressive applications in downstream  (Reviewers `RAbZ`, `Ut5k`, `jB9q`, `UWnW`).

In response to your valuable suggestions, we have conducted additional experiments and made the following modifications in the Rebuttal-PDF for your convenience:

- **About motivation**:
  - We incorporated additional literature to highlight Mamba's limitations in ICL in Section 2.2, as suggested by Reviewer `RAbZ`.

- **About baselines and performance**:
  - We included results for LongVA (suggested by Reviewers `RAbZ` and `jB9q`), showcasing our models' full performance potential in Table 2 (suggested by Reviewers `RAbZ` and `jB9q`), and provided parameter details for reference in Table 2,3,10 (suggested by Reviewer `jB9q`).

- **About ablations**:
  - **On staged training (Table 4) and replay data (Tables 13-15)**: An ablation study on staged training was added, following the suggestions of Reviewers `RAbZ` and `UWnW`. And ablation studies on replay data were incorporated, following the suggestions of Reviewer `jB9q`.
  - **More fair comparative experiment across architectures (Table 5, Tables 11-12)**: A new, more robust baseline for the Mamba Architecture was introduced in table 5 (suggested by Reviewer `RAbZ`). We also introduced an ablation analysis of different architectures with similar performance in the text to assess their impact on VLM, as recommended by Reviewer `jB9q`.
  - **On getting Jamba-9B dense model (Table 9)**: We included an ablation study on the strategies for obtaining dense hybrid models, as proposed by Reviewer `jB9q`.

- **A new Application**:
  - **Many-Shot ICL (Sec 6.3)**: Add results demonstrating the model's many-shot ICL capability to show models' potential for tasks where data is scarce or frequent updates are needed (suggested by Reviewer `UWnW`).

**If you have any additional questions or require further clarification, please feel free to let us know. Your insights are highly valued.**


Best regards,

The Authors

---

### Author Response · Authors · 2024-12-02

Dear Reviewers, Area Chair, and Program Chair,

As the review discussion period is drawing to a close (2nd Dec AoE for reviewer responses), we humbly request your reevaluation of our work based on our submitted responses. Your expertise and feedback are incredibly valuable in enhancing our research. We eagerly await your further comments and stand ready to address any additional concerns or questions you may have. We deeply appreciate your time and commitment to this rigorous process.

Warm Regards,
The authors of Paper 10001

---

### Meta-Review · Area_Chair_ozW9 · 2024-12-22

**Metareview:**

Summary: The paper introduces LongLLaVA, a novel multimodal large language model designed to enhance long-context capabilities. It employs a hybrid architecture combining Mamba and Transformer layers and uses token compression to efficiently process a large number of images. LongLLaVA demonstrates strong performance in multi-modal long-context understanding tasks and shows potential in healthcare and science applications.

Strengths:

LongLLaVA's hybrid architecture effectively balances computational efficiency with performance, allowing it to handle a large number of images on a single GPU.

The model shows promising results in long-context understanding tasks and has potential applications in various domains, indicating its versatility and practical value.

Drawbacks:

The paper lacks a clear justification for choosing the hybrid architecture over other potential solutions, which may affect the perceived novelty and necessity of the approach.

The performance comparison with other models is not entirely convincing, with some results showing significant degradation, raising questions about the model's robustness.

There is a need for a more comprehensive evaluation, including comparisons with more recent and competitive baselines, as well as a deeper analysis of the model's performance on various subtasks.

Given the above points, I must reject this work as it does not fully meet the acceptance criteria due to concerns about the clarity of its contributions, the completeness of performance evaluations, and the need for a more rigorous comparison with existing models.

**Additional Comments On Reviewer Discussion:**

Concerns are not well addressed.

---

### Decision · Program_Chairs · 2025-01-22

Reject